# Flat-surface-assisted and self-regulated oxidation resistance of Cu(111)

Su Jae Kim[1,12], Yong In Kim[2,12], Bipin Lamichhane[3,12], Young-Hoon Kim[2], Yousil Lee[1], Chae Ryong Cho[4], Miyeon Cheon[1], Jong Chan Kim[5], Hu Young Jeong[6], Taewoo Ha[7], Jungdae Kim[8], Young Hee Lee[2,7,9], Seong-Gon Kim[3 ✉], Young-Min Kim[2,7 ✉] & Se-Young Jeong[10,11 ✉]

Oxidation can deteriorate the properties of copper that are critical for its use, particularly in the semiconductor industry and electro-optics applications[1–7]. This has prompted numerous studies exploring copper oxidation and possible passivation strategies[8]. In situ observations have, for example, shown that oxidation involves stepped surfaces: $Cu_2O$ growth occurs on flat surfaces as a result of Cu adatoms detaching from steps and diffusing across terraces[9–11]. But even though this mechanism explains why single-crystalline copper is more resistant to oxidation than polycrystalline copper, the fact that flat copper surfaces can be free of oxidation has not been explored further. Here we report the fabrication of copper thin films that are semi-permanently oxidation resistant because they consist of flat surfaces with only occasional mono-atomic steps. First-principles calculations confirm that mono-atomic step edges are as impervious to oxygen as flat surfaces and that surface adsorption of O atoms is suppressed once an oxygen face-centred cubic (fcc) surface site coverage of 50% has been reached. These combined effects explain the exceptional oxidation resistance of ultraflat Cu surfaces.

Given that the step edge is vulnerable to oxidation because surface steps act as the dominant source of Cu adatoms for oxide growth on surface terraces[2,11], oxidation resistance requires that surface step edges are avoided[6,12,13]. In this regard, the close-packed Cu(111) surface is superior to other Cu surfaces[14,15] and our experimental demonstration thus used a single-crystal Cu(111) film (SCCF) grown by atomic sputtering epitaxy (ASE)[16] to show that a tightly coordinated flat surface can remain semi-permanently stable against oxidation. Theoretical calculations show the atomic-scale oxidation-resistant mechanism of the flat copper surface from the perspective of the likely pathways for oxygen atoms into the viable structures of the copper surface with a discovery of the self-regulated protection layer at elevated oxygen coverages. The implication is that the atomically flat surface of the SCCF shows oxidation-resistant properties owing to the high energy barrier for oxygen infiltration and self-regulation owing to high oxygen coverages.

## Flatness of Cu(111) surface

The surface and structural characteristics of a 110-nm-thick SCCF with an ultraflat surface (see Extended Data Fig. 1 for a large-scale characterization) are examined using high-resolution (scanning) transmission electron microscopy (HR(S)TEM) combined with geometrical phase analysis (GPA)[17,18] (Fig. 1). The cross-sectional (S)TEM images (Fig. 1a, e) show that the copper film grew along the [111] direction, thus creating an exposed surface (111) plane with mono-atomic step-edge structures. Typical multi-atomic step edges and intrinsic defects such as grain boundaries and stacking faults are rarely detected. It is remarkable that the outermost copper surface layer has the same atomic configuration as the interior copper without evidence of surface relaxation or structural changes by surface oxidation, even at the step-edge positions. To examine the local strain behaviour near the surface region, lattice distortions along the in-plane ($x$) and out-of-plane ($y$) directions relative to the inside of the SCCF were measured by the GPA technique (Fig. 1b, c). The resulting strain field maps ($E_{xx}$ and $E_{yy}$) clearly show that no noticeable change in lattice strain is observed throughout the entire surface region. This means that the SCCF has a nearly perfect atomic structure up to its outermost surface layer without any structural defects, such as vacancies or dislocations. The simulated HRTEM image using an amorphous carbon/flat copper surface model matches well with the experimental HRTEM image (Fig. 1a). By comparing the layer spacings of the (111) stacking planes ($d_{(111)} = 0.21$ nm) between the simulated and experimental images (Fig. 1d), it is evident that the Cu surface is undistorted and ultraflat, and has the same structure as bulk Cu. Annular dark-field (ADF) and annular bright-field (ABF) STEM images of the SCCF surface

[1]Crystal Bank Research Institute, Pusan National University, Busan, Republic of Korea. [2]Department of Energy Science, Sungkyunkwan University, Suwon, Republic of Korea. [3]Department of Physics and Astronomy, Mississippi State University, Mississippi State, MS, USA. [4]Department of Nanoenergy Engineering, Pusan National University, Busan, Republic of Korea. [5]School of Materials Science and Engineering, Ulsan National Institute of Science and Technology, Ulsan, South Korea. [6]UNIST Central Research Facilities (UCRF), Ulsan National Institute of Science and Technology, Ulsan, South Korea. [7]Center for Integrated Nanostructure Physics, Institute for Basic Science (IBS), Sungkyunkwan University, Suwon, Republic of Korea. [8]Department of Physics, University of Ulsan, Ulsan, Republic of Korea. [9]Department of Physics, Sungkyunkwan University, Suwon, Republic of Korea. [10]Department of Optics and Mechatronics Engineering, Pusan National University, Busan, Republic of Korea. [11]Department of Cogno-Mechatronics Engineering, Pusan National University, Busan, Republic of Korea. [12]These authors contributed equally: Su Jae Kim, Yong In Kim, Bipin Lamichhane. ✉e-mail: sk162@msstate.edu; youngmk@skku.edu; syjeong@pusan.ac.kr

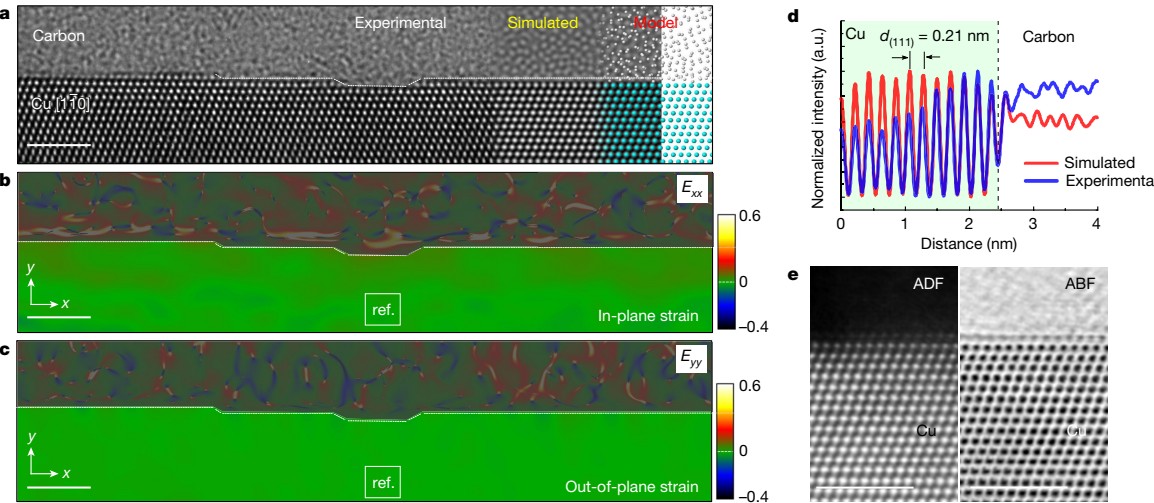

**Fig. 1 | Surface of a single-crystal copper thin film grown by ASE.**
**a**, Cross-sectional HRTEM image of the surface region of the copper thin film
observed in the [1$\bar{1}$0] orientation. The simulated HRTEM image and the
corresponding model of the carbon–copper supercell are presented alongside
the experimental HRTEM image. Scale bar is 2 nm. **b**, **c**, In-plane ($E_{xx}$) and
out-of-plane ($E_{yy}$) strain field maps obtained by GPA for the experimental
HRTEM image. The reference area of Cu chosen for GPA is marked by the white

square boxes in each map. Note that complex patterns for parts of the carbon
film are shaded by a grey colour for clarity, as those are not relevant for the
strain behaviour of the SCCF. Scale bars are 2 nm. **d**, Comparison of the
experimental and simulated intensity profiles obtained along the out-of-plane
direction for (111) layer spacings ($d_{(111)} = 0.21$ nm). **e**, Simultaneously acquired
ADF-STEM and ABF-STEM images for the surface region of the copper thin film.
Scale bars are 2 nm.

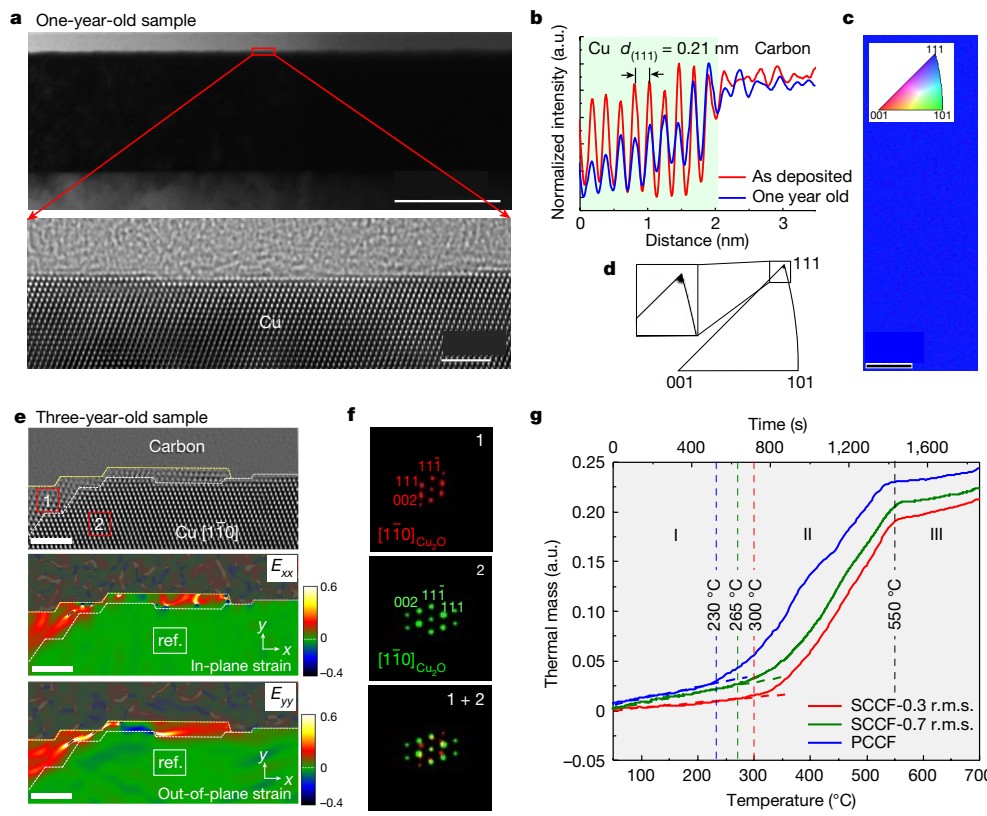

**Fig. 2 | Long-term oxidation resistance and structural stability of the SCCF
surface. a**, Low-magnification BF-TEM image of the SCCF sample exposed to
ambient air at room temperature for about 1 year (top). The result shows that
the atomically flat surface morphology over the entire SCCF film has remained
almost unchanged. HRTEM image of the surface region (marked by a red
rectangle in the top panel) of the 1-year-old SCCF sample (bottom). The sample
was oriented in the [1$\bar{1}$0] direction. Scale bars are 100 nm (top) and 2 nm
(bottom). **b**, Comparison of the intensity profiles for the (111) plane between
the as-deposited (red) and 1-year-old (blue) samples. **c**, EBSD map showing

perfect alignment along the (111) plane. Scale bar is 3 μm. **d**, IPF with a sole spot
associated with the (111) plane. The inset image is the enlarged image of the
sole-spot area. **e**, HRTEM image of the surface region of the 3-year-old SCCF
sample observed in the [1$\bar{1}$0] orientation (top) and in-plane ($E_{xx}$, middle) and
out-of-plane ($E_{yy}$, bottom) strain field maps obtained by GPA for the HRTEM
image of the 3-year-old SCCF sample. Scale bars are 2 nm. **f**, FFT patterns of
region 1 (top), region 2 (middle) and both regions (bottom).
**g**, Thermogravimetric analysis for the PCCF and SCCF samples with different
surface roughness. r.m.s., root mean square.

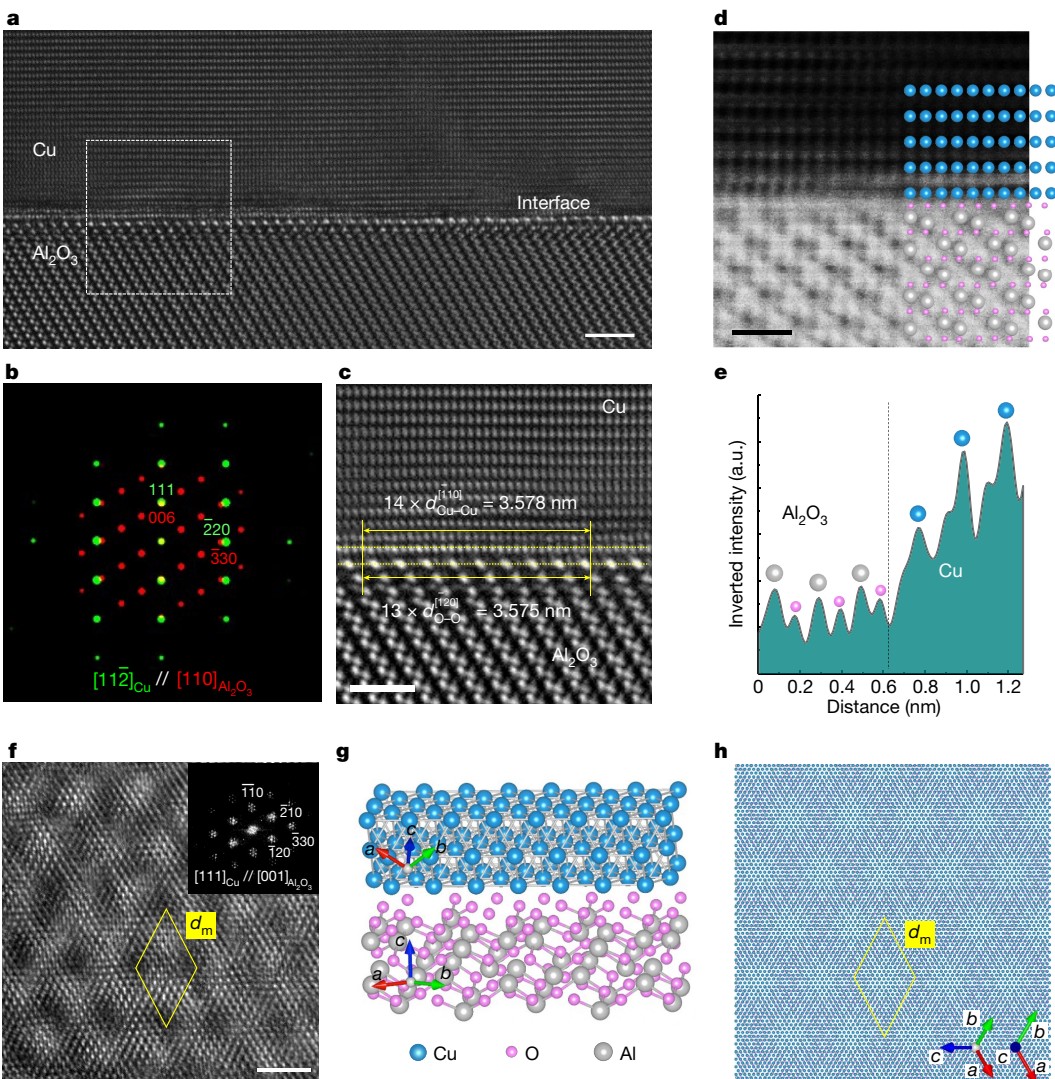

**Fig. 3 | Interface structure and crystallographic relation. a**, Cross-sectional HRTEM image of the Cu–Al$_2$O$_3$ heterointerface having the orientation relation of $(111)_{Cu}[11\bar{2}]_{Cu}//(001)_{Al_2O_3}[110]_{Al_2O_3}$. Scale bar is 2 nm. **b**, A composite pattern of FFTs for the regions of Cu and Al$_2$O$_3$ in the HRTEM image. **c**, Enlarged HRTEM image for the region marked by the dashed box in **a**. Scale bar is 1 nm. **d**, ABF-STEM image and superimposed atomic model of the Cu–Al$_2$O$_3$ heterointerface. Scale bar is 5 Å. **e**, Inverted intensity profile obtained across the interface between Cu and Al$_2$O$_3$ in **d**. **f**, Plan-view HRTEM image of the Cu–Al$_2$O$_3$ heterointerface. The inset is the FFT pattern of the HRTEM image. Scale bar is 2 nm. **g**, **h**, Side view (**g**) and plan view (**h**) of the deduced epitaxy model of Cu grown on an Al$_2$O$_3$ substrate. The yellow diamonds with unit length of $d_m$ in **f** and **h** indicate the 2D unit cell of the moiré pattern.

are complementary to the HRTEM observation results (Fig. 1e). We can discriminate this remarkable ultraflat surface structure of the SCCF from that of the conventional polycrystalline Cu thin film (PCCF) and the Cu(111) surface of a bulk Cu single crystal with a notable oxidized copper layer on top of them (Extended Data Fig. 2 and Fig. 3). Notably, we found that the SCCF sample maintained its ultraflat and pristine surface even after more than a year of air exposure (Fig. 2a–d), suggesting that our SCCF has exceptional oxidation-resistant properties (also see Extended Data Fig. 4 for a large-scale scanning tunnelling microscope (STM) topography).

The low-magnification bright-field TEM (BF-TEM) image (Fig. 2a, top) and the cross-sectional HRTEM image (Fig. 2a, bottom) show that the mono-atomic step-edge structure is maintained even after a year. The two intensity profiles for the (111) plane between the as-deposited (red) and the 1-year-old (blue) samples (Fig. 2b) show a change in the planar spacing of the (111) planes up to the uppermost surface layer, and they agree well with each other, suggesting that no notable oxidation has occurred even in the SCCF under long-term exposure to air. The electron backscatter diffraction (EBSD) map

(Fig. 2c) and inverse pole figure (IPF) (Fig. 2d) show that no misalignment of crystal lattices deviating from the (111) plane has occurred after a year. Oxidized portions were rare on the 3-year-old SCCF sample oriented along [1$\bar{1}$0] in the HRTEM image, although they were found on the sample edge where the sample was cut (Fig. 2e, top). The resulting strain field maps ($E_{xx}$ and $E_{yy}$) prepared using the GPA technique (Fig. 2e, middle and bottom) show that the overlayer (region 1) has a new lattice structure that is mismatched with the SCCF. The fast Fourier transform (FFT) patterns of region 1 (Fig. 2f, top), region 2 (Fig. 2f, middle) and both regions (Fig. 2f, bottom) indicate that the two regions belong to the Cu$_2$O and Cu phases, respectively, and suggest that part of the surface is oxidized. However, the oxidized surface only reaches a few Cu$_2$O layers, which is thin compared with the thickness of natural oxide layers in polycrystalline Cu (Extended Data Fig. 2). Comparative thermogravimetric analysis of PCCF and SCCF samples having different surface roughness clearly demonstrates that the SCCF with mono-atomic step edges shows exceptional resistance to initial oxidation at elevated temperatures compared with the other samples (Fig. 2g).

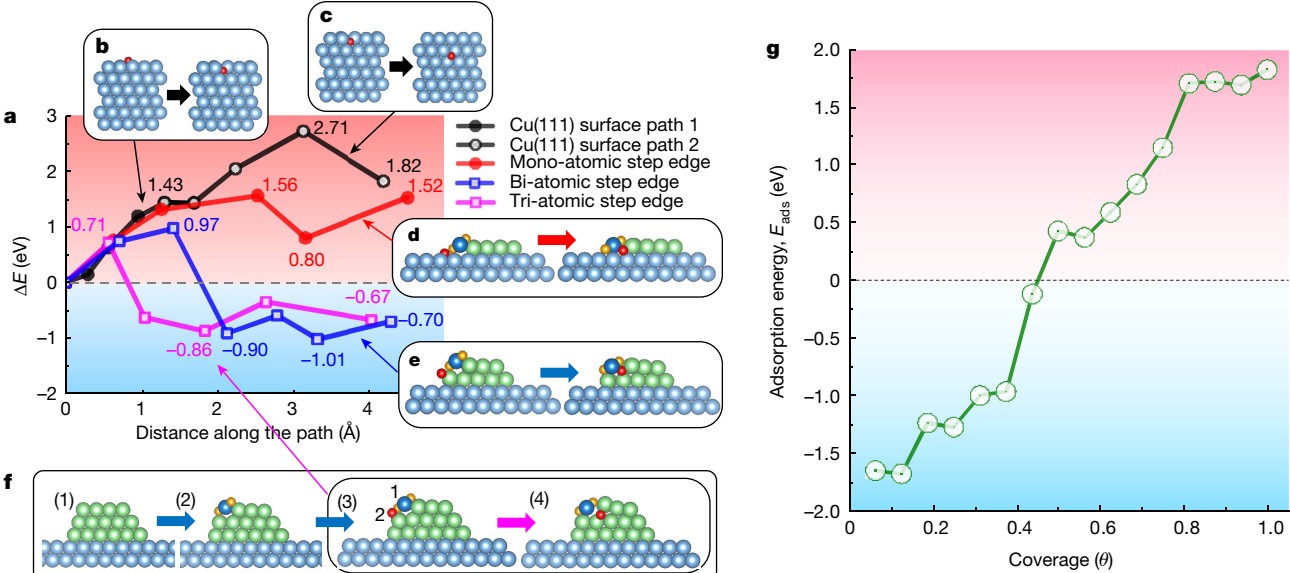

**Fig. 4 | Theoretical analysis and a model of oxidation of the copper surface.** **a**–**f**, Energy profile of an O atom along various penetration paths (**a**): from the Cu(111) surface into the first Cu substrate interstitial layer (black solid circles) (**b**); from the first to the second Cu substrate interstitial layer (black open circles) (**c**); from the outside to the inside of the mono-atomic step edge (red solid circles) (**d**); from the outside to the inside of the bi-atomic step edge (blue open squares) (**e**); and from the outside to the inside of the tri-atomic step edge (pink open squares) (**f**). Blue spheres represent Cu atoms in the bulk or substrate, green spheres represent Cu atoms in the steps and dark blue spheres represent Cu atoms on the step edge. Orange spheres represent adsorbed O atoms and red spheres represent O atoms infiltrating into the interstitial region. Panel **f** also shows the initiation of oxidation on the edge of a

multi-atomic step: (1) given a pristine step edge and ambient oxygen conditions, (2) two O atoms (orange spheres) adsorb on each side of the Cu on the edge (Cu-1, dark blue sphere); (3) the third adsorbed O atom (O-2, red sphere) causes the local structure of Cu-1 to be similar to that of Cu in a monolayer of Cu$_2$O, and the expansion of volume causes Cu-1 to move upward and open a pathway for O-2; and (4) O-2 passes through the opening and binds with Cu atoms in the next row to push those Cu atoms upward to sustain the oxidation process. **g**, Incremental adsorption energy of O atom as a function of oxygen coverage of the fcc sites on the Cu(111) surface. The red and blue shading in **a** and **g** represents the endothermic and exothermic reactions, respectively.

## Growth condition for the flat surface

The flatness of a surface is decisively influenced by the interface structure between the film and the substrate[19,20], which can be relaxed by structural defects such as dislocations. The interface structure between the Cu film and the Al$_2$O$_3$ substrate is characterized by HR(S)TEM (Fig. 3). The overall interface structure viewed at the [11$\bar{2}$] orientation of the Cu film and the FFT pattern of the image show that the crystallographic orientation relationship (OR) is (111)$_{Cu}$[11$\bar{2}$]$_{Cu}$//(001)$_{Al_2O_3}$[110]$_{Al_2O_3}$ (Fig. 3a, b). The copper lattices seem to flawlessly adjoin with the Al$_2$O$_3$ substrate without interfacial misfit defects, suggesting that the Cu film grows metamorphically on the substrate[21] (Extended Data Fig. 5). The enlarged image of the interface (Fig. 3c) shows a detailed lattice mismatch between the two materials. The in-plane atomic distance mismatch $f((d_{O-O_{Al_2O_3}}^{[\bar{1}20]} - d_{Cu-Cu}^{[\bar{1}10]})/d_{O-O_{Al_2O_3}}^{[\bar{1}20]})$ between Cu atoms in the film and O atoms in Al$_2$O$_3$ is estimated to be 6.9%. However, considering the extended atomic distance mismatch (EADM)[22,23], the large mechanical misfit strain can be relieved if the mismatch for a relatively long period of atomic distance is extremely small. The EADM is defined as $(ID - I'D')/I'D'$, in which $D$ and $D'$ are the Cu–Cu distance in the Cu (111) epilayer and the O–O distance in the substrate, respectively, and $I$ and $I'$ are smallest non-reducible integers determined by the relation $D{:}D' \sim I{:}I'$. Given the $D_{Cu-Cu}$ (14 × $d_{Cu-Cu}^{[\bar{1}10]}$) interatomic spacing (3.578 nm) of Cu atoms in the film and the $D'_{O-O}$ (13 × $d_{O-O}^{[\bar{1}20]}$) interatomic spacing (3.575 nm) of O atoms in Al$_2$O$_3$, the EADM of the Cu–Al$_2$O$_3$ interface is about 0.1% (Fig. 3c). The light-element-sensitive ABF-STEM imaging[24] (Fig. 3d) shows that the interface model of Cu grown on the oxygen-terminated Al$_2$O$_3$ surface matches well with the experimental heterostructure. The inverted intensity profile obtained across the interface (Fig. 3e) clearly cor-

roborates the presence of an oxygen layer between the Cu and Al layers, thus indicating the existence of Cu–O interactions at the interface, which can be stabilized on a typical Al$_2$O$_3$ surface terminated with oxygen[25]. Owing to the in-plane lattice mismatch, a large-scale interference pattern, that is, a moiré pattern[26], can be observed in the plane view of the Cu–Al$_2$O$_3$ heterostructure (Fig. 3f). Indeed, the hexagonal moiré pattern with a dimension ($d_m$) of 1.83 nm is observed owing to the different in-plane lattice periodicities in the vertical OR of [111]$_{Cu}$//[001]Al$_2$O$_3$, which is confirmed by the FFT pattern analysis (inset in Fig. 3f). The simulated moiré pattern generated by the epitaxy model of Cu–Al$_2$O$_3$ with the same OR is consistent with the experimental moiré pattern, showing the repeating large-scale contrast feature (Fig. 3g, h). This vertical OR observation corroborates our EADM analysis and indicates that the growth mechanism of the SCCF on the Al$_2$O$_3$ substrate can be understood on the basis of the large-scale mismatch epitaxial relationship, rather than the atomic-scale lattice interrelation. The detailed chemical nature of the SCCF at the surface and the interface regions are investigated by the combined spectroscopic approaches of energy-dispersive X-ray spectroscopy (EDX), electron energy loss spectroscopy (EELS) in ADF-STEM imaging mode and X-ray photoelectron spectroscopy[27,28] (Extended Data Fig. 6).

## Calculation of oxidation resistance

We can understand the exceptional oxidation resistance of our films using a microscopic model of copper oxidation on the basis of first-principles density functional theory (DFT) calculations[29–31]. The main reason for the suppression of oxidation is that our atomically flat film is free of one critical feature, namely, multi-atomic step edges, as shown in Fig. 4. The energy profile in Fig. 4a shows that the penetration of an O atom through

a pristine Cu(111) surface needs an activation energy of more than 1.4 eV (Fig. 4b), and the O atom becomes stable only in the second subsurface interlayer space with a further energy barrier of 1.3 eV (Fig. 4c). One of the main reasons for the strong resistance of a flat surface to oxidation is that, when Cu atoms are oxidized, the out-of-plane distance between Cu layers increases from 2.10 to 2.48 Å, and the volume expands by 18% (Extended Data Fig. 7). The volume increase is even more pronounced in the early stage of oxidation: the Cu layer distance in the monolayer of $Cu_2O$ on the Cu surface is 3.26 Å, and the volume is greater by 55% (Extended Data Fig. 7). Given that one exposed Cu surface is insufficient to initiate the oxidation process, we examine a structure wherein two crystallographically different planes meet, namely, the edge of a multi-atomic step. Notably $O_2$ molecules physisorbed on a Cu(111) surface readily dissociate into O ions with a small activation energy of 0.027 eV (Extended Data Fig. 8). Figure 4d–f shows the initiation of oxidation on the edge of such multi-atomic steps according to our DFT calculation; Fig. 4a shows that there is a critical difference between a mono-atomic step and a multi-atomic step in terms of the oxidation resistance. Whereas the penetration of O atoms at a bi-atomic and tri-atomic step is an exothermic reaction ($\Delta E = -0.90$ and $-0.86$ eV, respectively) with smaller activation energies ($E_a = 0.97$ and $0.71$ eV, respectively), the penetration of O at the edge of a mono-atomic step is highly endothermic ($\Delta E = +0.80$ eV), requiring a much larger activation energy ($E_a = 1.56$ eV), and shows strong oxidation resistance. Our DFT calculation indicates that, as the coverage of oxygen increases, the incremental adsorption energy for the next O atom becomes smaller and eventually becomes negative beyond 50% of oxygen coverage, rendering the adsorption process energetically unfavourable (Fig. 4g). This self-regulation of oxygen suppresses further adsorption of O atoms on an ultraflat Cu surface at elevated oxygen coverages and increasingly enhances the oxidation resistance of the surface. Our microscopic model of oxidation is well supported by HRTEM images, which show that atomically flat Cu thin films with occasional presence of mono-atomic steps are highly resistant to oxidation over an extended period (≥3 years) (Fig. 2), whereas Cu films with multi-atomic steps show substantial oxidation (Extended Data Figs. 2 and 3). The oxidation resistance of a Cu(111) surface is greatly affected by the type of surface defects[32–35], which suggests that an atomically flat Cu(111) surface without multi-atom steps is essential to achieve strong oxidation resistance.

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

# Methods

## ASE for the preparation of single-crystal Cu thin films

Cu thin films were grown as nearly defect-free and grain-boundary-free single crystals using the ASE technique achieved by addressing the problems of conventional sputter systems[16]. Because the ASE system controlling atomic-level growth is very sensitive to environmental factors, it requires three key instrumental modifications compared with a commercially available sputtering system. First, we used a single-crystal target instead of a polycrystalline one; second, all conducting wires of the wiring network were replaced with single-crystal copper wires; finally, a mechanical noise-reduction system was installed to suppress the mechanical vibrations from the surroundings. The idea of this technology is on the basis of completely eliminating the noise caused by electron-grain boundary scattering in the conduction network in the device, which, to our knowledge, has been completely ignored until now. Once the system was appropriately constructed, we confirmed that the quality of the Cu films was greatly improved, with a high level of reproducibility. The schematic diagram of the ASE system with the three key features is shown in Extended Data Fig. 9a and the detailed descriptions are as follows.

**Single-crystal Cu sputtering target.** For a polycrystalline target with a high density of surface steps and grain boundaries, atoms on the edges or grain boundaries have a lower binding energy than that of atoms arranged on a flat plane of the grain. Hence the atoms at the structural defects are prone to being sputtered as atomic clusters for the RF power set to remove atoms from a flat plane, which will eventually be deposited on the substrate as randomly oriented clusters. To realize the practicability of ASE growth, the use of a single-crystal Cu target with a (111) surface is essential because Cu atoms are sputtered from the target as individual atoms. Thus the uniform stacking of Cu atoms on the substrate is empirically achieved for the growth of an ultraflat film. The single-crystal Cu(111) target can be obtained from the single-crystal Cu ingot grown by the Czochralski method by cutting using wire electrical discharge machining as a 6-mm-thick disc with a 2-inch diameter (see the two images on the upper-left side of Extended Data Fig. 9a). Even though Cu single-crystal ingots are commercially available, we grew them using our own apparatus in this study, and there were no differences between the two in the resulting improvement of film quality.

**Electrical noise reduction using single-crystal Cu wires.** To reduce electrical noise interference, we replaced the electrical networks made of conducting wires in conventional sputtering systems with single-crystal Cu wires as much as possible (see the three images on the lower-left side of Extended Data Fig. 9a). The effectiveness of this modification was previously demonstrated in a Hall measurement kit with circuitry and connecting components made of single-crystal Cu, notably improving the measurement precision of the electrical coefficients, such as carrier density and mobility[36]. Single-crystal Cu wires can be prepared by the wire electrical discharge machining cutting of a single-crystal Cu disc in a spiral manner, as reported in our previous study[37]. To ensure further reduction of electrical noise, we replaced a typical RF power cable with a single crystal. To monitor how effectively the single-crystal power cable improves the RF power stability in the sputtering system, we measured the change in the RF power over time with and without the single-crystal Cu power cable (Extended Data Fig. 10a, b). Single-crystal wiring has already been adapted in the sputtering system used in the test. Therefore it is evident that the output RF power is more stabilized in a narrower power range (12.61 ± 0.005 W) after changing the original power cable to a single crystal. Although the RF power stability of the sputtering system before modification is also of good quality compared with the conventional ones, we empirically confirmed that such a

level of stability is not sufficient for the growth of the Cu film with an atomic-level flatness.

**Mechanical noise reduction using mechanical diodes.** Although mechanical noise is not the main source of interference in the ASE system, the electrical noise reduction on the basis of the single-crystal Cu wiring cannot be effective unless the mechanical noise is effectively screened. After the tests to reduce the mechanical noise with several choices, including absorbers, barriers, vibration isolators and vibration dampers, we found that the application of a mechanical diode consisting of a set of metal spikes and pads is very effective and economical in shielding the mechanical interferences transmitted through the wall and the floor. As depicted in Extended Data Fig. 9a, we designed mechanical diodes and installed them on every device, including the chamber and the vacuum pumps. The growth of the atomically flat metal films was not successful without this shield against mechanical noise.

To verify the reproducibility of our ASE approach for the growth of the ultraflat Cu(111) film, we measured the root mean square (r.m.s.) roughness of many samples as a function of the film thickness and compared these values with those of the samples grown by a conventional sputtering system equipped with only the single-crystal Cu target (Extended Data Fig. 10c). The averaged r.m.s. value for the 29 samples grown by the ASE system was estimated to be around 0.20 ± 0.1 nm (red dashed line), which is similar to the planar spacing of Cu(111). Notably, it can be further decreased to about 0.17 ± 0.1 nm (purple dashed line) when considering thinner Cu films below 35 nm, suggesting their reliable applications to ultrathin electronic devices. However, the averaged r.m.s. value for the samples grown by the conventional sputtering system equipped with the single-crystal Cu target was around 0.66 ± 0.2 nm, which is good but not enough for the growth of the ultraflat Cu films with only the mono-atomic steps.

**Optimized sputtering conditions using the ASE system.** A double-side polished (001) $Al_2O_3$ wafer with a thickness of 430 μm was used as the substrate material. The optimized deposition temperature and RF (13.56 MHz in frequency) power were about 170 °C and 30 W, respectively, and varied slightly, depending on the ASE systems. The target-to-substrate distance was set at 95 mm. The base pressure was maintained at under $2 \times 10^{-7}$ torr and the working pressure at $5.4 \times 10^{-3}$ torr with an Ar gas flow of 50 sccm. Ar gas with a purity of 99.9999% (6N) was used. The relationship between the deposition time and the thickness of the thin film (or the average growth rate) was determined from the average deposition time of a 200-nm-thick film grown under the optimum conditions. The determined average growth rate of roughly 4.3 nm min$^{-1}$ is fairly reliable above a film thickness of 10 nm. The kinetic energy of sputtered Cu atoms depends on the incident ion energy of Ar$^+$ and the binding energy of Cu atoms at the surface of the target. The crystallographically different surfaces of Cu have different surface binding energies ($E_{b,Cu}$), which were reported to be 4.62, 4.26 and 4.65 eV for the Cu(100), Cu(110) and Cu(111) planes, respectively[38]. Considering the potential energy ($E_{Ar^+}$ = 15eV) of the accelerated Ar$^+$ ion at the maximum current of 1 A in our ASE equipment, the kinetic energy of the sputtered Cu atoms can be narrowly distributed at around 10.35 eV, which is roughly calculated from $E_{Ar^+} - E_{b, Cu(111)}$ in the case of the Cu(111) single-crystal target used in this study. By contrast, in the case of polycrystalline Cu target dominantly having a mixture of Cu(100), Cu(110) and Cu(111) exposed planes at the surface, the kinetic energy of the sputtered Cu atoms is expected to be spread between 10.35 and 10.74 eV. When considering the surface defects and grain boundaries at the target, the kinetic energy would be distributed more widely up to 11.52 eV (ref. [39]). The radial distribution of the incident flux of the sputtered Cu atoms at the substrate is known to determine the uniformity of the deposition thickness of Cu film[40]. To check the thickness uniformity of the grown SCCF, we measured the thickness at five different points from centre to edge in a 2-inch wafer using an atomic

force microscope (AFM). As a result, the thickness uniformity was estimated to be about 99.8% (Extended Data Fig. 9b, c). This result suggests that the diffusive flux of Cu atoms is purely uniform at the position of the substrate. Note that the substrate of our system is rotated at 30 rpm.

## Structural and chemical characterizations

X-ray diffraction $\theta$–$2\theta$ measurements were performed using a PANalytical Empyrean Series 2 instrument equipped with a Cu $K\alpha$ source (40 kV, 30 mA). Data were collected in the range $20° < 2\theta < 90°$, with a step size of 0.0167° and a dwell time of 0.5 s per point in all cases. AFM measurements were carried out using an XE-100 system (Park Systems, Inc.). Scanning electron microscopy, EBSD, pole figure (PF) and IPF measurements were performed with a Zeiss Supra 40VP with a scanning electron microprobe. An STM surface analysis was conducted using a custom-built STM system installed at the University of Ulsan, Korea. Electron-transparent cross-sectional TEM samples were prepared by the Ga ion beam milling and lift-out method in focused ion beam systems (FIB, Helios NanoLab 450, FEI and AURIGA CrossBeam Workstation, Carl Zeiss) and the possible damaged surface layers on the samples were removed by subsequent low-energy Ar ion beam surface milling at 700 eV for 15 min (Model 1040 NanoMill, Fischione). The plan-view TEM specimen was prepared by mechanical polishing and dimple grinding, followed by ion milling with Ar ions. Double $C_s$-corrected (S)TEM systems (JEM-ARM200CF, JEOL) equipped with EELS (Quantum ER965, Gatan) and EDX (JED-2300T, JEOL) were used for atomic-scale structure imaging and chemical analysis of the samples at an accelerating voltage of 200 kV. The inner and outer angle ranges for ADF-STEM and ABF-STEM imaging were 45–180 and 12–24 mrad, respectively. The HRTEM, ADF-STEM and ABF-STEM images were denoised by a local 2D difference image filter that is implemented in commercial software (HREM-Filters Pro, HREM Research Inc.). The HRTEM simulation was carried out for the amorphous carbon/copper [$1\bar{1}0$] supercell structure ($4.8 \times 4.8$ nm$^2$) using the multislice method, which is implemented in the commercial software MacTempas (Total Resolution LLC), with the following microscope and imaging parameters: accelerating voltage ($V = 200$ kV), spherical aberration coefficient ($C_s = 0.4$ μm), chromatic aberration coefficient ($C_c = 1.1$ mm), convergence semi-angle ($\alpha = 0.5$ mrad), sample thickness ($t = 20.5$ nm) and defocus ($\Delta f = +14$ nm). The simulated HRTEM image was estimated to have a correlation of 0.98 with the experimental HRTEM image as a cross-correlation factor. For the quantitative analysis of local strain components in the copper thin film, the GPA technique was used, which allows mapping two-dimensional local displacement fields by analysing the phase shift between non-collinear Fourier components of the lattice vectors $g_1$ and $g_2$. For EELS measurements of the Cu–Al$_2$O$_3$ interface and the Cu surface, the core-loss EELS spectra of the Cu $L$ edge were obtained from the interface to the Cu surface using the line scan function of the scanning step (0.72 nm) for a 37.83-nm length with an energy dispersion of 0.5 eV pix$^{-1}$ and a dwell time of 2.0 s pix$^{-1}$. For core-loss EELS spectrum imaging for the comparison of surface structures between the SCCF and the PCCF samples, surface regions of both samples were selected by $20 \times 16$ pixels that can be translated as $9.87 \times 7.9$ nm$^2$ and scanned with the step size of 0.493 nm to obtain EELS spectrum imaging dataset. The selected range of energy loss was set to be 477–988 eV including O $K$ and Cu $L_{2,3}$ edges. Nanoscale STEM-EDX maps of the constituent elements of the Cu–Al$_2$O heterostructure were obtained for a $256 \times 256$ pixel resolution with a high-efficiency dual silicon drift detector X-ray detector system having a wide collection window of 100 mm$^2$ for each detector, and the sample drift during the acquisition was corrected by tracking the reference area assigned at the acquisition setup.

## Thermogravimetric analysis for the PCCF and SCCF samples with different surface roughness

The thermal mass change was measured using a thermogravimetric measurement system (TG-DTA 2000S, MAC Science). We prepared two SCCF samples with different values of surface roughness and a PCCF sample. The gravimetric changes of the three samples were measured in the temperature range from room temperature to 700 °C at a heating rate of 20 °C min$^{-1}$ under air atmosphere. The two SCCF samples with different values of r.m.s. roughness, that is, 0.3 nm (SCCF-0.3, corresponding to a mono-atomic step) and 0.7 nm (SCCF-0.7, corresponding to a bi-atomic or tri-atomic step), were tested to ascertain the effect of the surface steps on oxidation in comparison with the PCCF sample (blue) (r.m.s. roughness around 10 nm). Although the SCCF sample with a roughness of 0.3 nm (red) was synthesized by ASE and had an ultraflat surface with occasional mono-atomic steps, the SCCF sample with a roughness of 0.7 nm (green), which was synthesized using a single-crystal target in a conventional sputtering system, had occasional multi-atomic steps of 2 or 3 atomic layers in height.

## Theoretical calculations

All ab initio total energy calculations and geometry optimizations were performed with DFT in the generalized gradient approximation Perdew–Burke–Ernzerhof functional[29] and with the projected augmented-plane-wave method[30], as implemented by Kresse and Joubert[31]. The Cu substrate was represented by slabs of six layers with the theoretical equilibrium lattice constant. A vacuum length of 15 Å was used, and the bottom two layers of the slab were fixed in their bulk positions. The electron wave functions were expanded in a plane-wave basis set with a cut-off energy of 400 eV. The Brillouin zone for the slabs was sampled using $k$-point sets equivalent to at least a ($5 \times 5 \times 1$) Monkhorst–Pack grid[41] for the primitive fcc unit cell. The climbing image-nudged elastic band method[42] was used to calculate activation energies. The local atomic charge was computed using Bader's charge decomposition method[43], which divides the total volume into individual atomic volumes for each atom as the one containing a single charge density maximum and separated from the other volumes by a zero-flux surface of the gradients of the charge density magnitude. The incremental adsorption energy of an oxygen atom as a function of oxygen coverage $\Theta = n/n_0$ for $n$ O atoms on $n_0$ fcc sites on a Cu(111) substrate is calculated as the energy change when a further O atom is adsorbed on the substrate, $E_{ad}(n) = E[Cu + (n+1)O] - E[Cu + nO] - \frac{1}{2}E[O_2]$, in which $E[Cu + nO]$ is the total energy of $n$ oxygen atoms on the Cu substrate and $E[O_2]$ is the total energy of an isolated oxygen molecule.

## Data availability

The data that support the findings of this study are available from the corresponding authors on reasonable request. Source data are provided with this paper.

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

**Acknowledgements** This research was supported by the Basic Science Research Program through the National Research Foundation of Korea (NRF) funded by the Ministry of Science, ICT and Future Planning (nos. NRF-2017R1A2B3011822 and NRF-2020R1A4A4078780), the Creative Materials Discovery Program (nos. NRF-2015M3D1A1070672 and NRF-2016M3D1A1919181) through the NRF, and the Institute for Basic Science (IBS-R011-D1). Computer time allocation was provided by the High Performance Computing Center (HPCC) at Mississippi State University.

**Author contributions** S.-Y.J., Y.-M.K. and S.-G.K. conceived this work and wrote the manuscript. S.-Y.J. and Y.-M.K. supervised the experiments. S.J.K. and Y.L. prepared the SCCF samples. Y.-M.K., Y.I.K., Y.-H.K., J.C.K. and H.Y.J. performed TEM analyses and sample preparations. J.K. conducted the STM analysis. S.J.K., Y.L., T.H., C.R.C. and M.C. performed EBSD and AFM experiments and assisted with the data analyses. S.-Y.J. and S.-G.K. established the theoretical model, and S.-G.K. and B.L. carried out first-principles calculations. S.-Y.J. and Y.H.L. supervised and coordinated this work. All authors contributed to the discussion and analysis of the results.

**Competing interests** The authors declare no competing interests.

**Additional information**
**Correspondence and requests for materials** should be addressed to Seong-Gon Kim, Young-Min Kim or Se-Young Jeong.

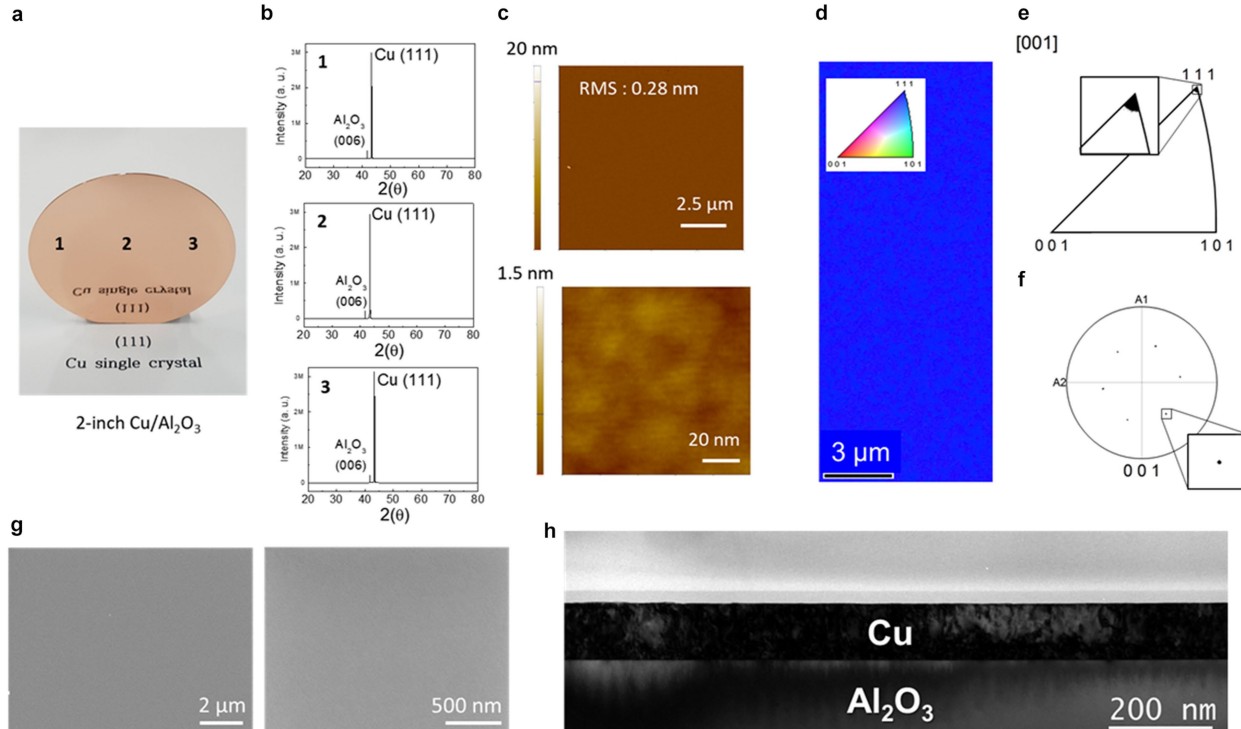

**Extended Data Fig. 1 | Pristine SCCFs grown by ASE. a**, Photo of a 2-inch SCCF grown on $Al_2O_3$. **b**, $\theta$–$2\theta$ X-ray diffraction data taken at different positions marked in **a**. **c**, Surface morphologies at 20-nm (upper) and 1.5-nm (lower) vertical scales of AFM images with a root mean square (RMS) surface roughness of 0.28 nm. **d**, EBSD map showing perfect alignment along the (111) plane. **e**, IPF with a sole spot associated with the (111) plane. **f**, [100] PF showing the sixfold symmetry of the {100} PF. The inset images in **e** and **f** are the enlarged images of the sole-spot areas. **g**, Scanning electron microscopy images of the sample at different magnifications. **h**, Low-magnification cross-sectional BF-TEM image of the SCCF sample.

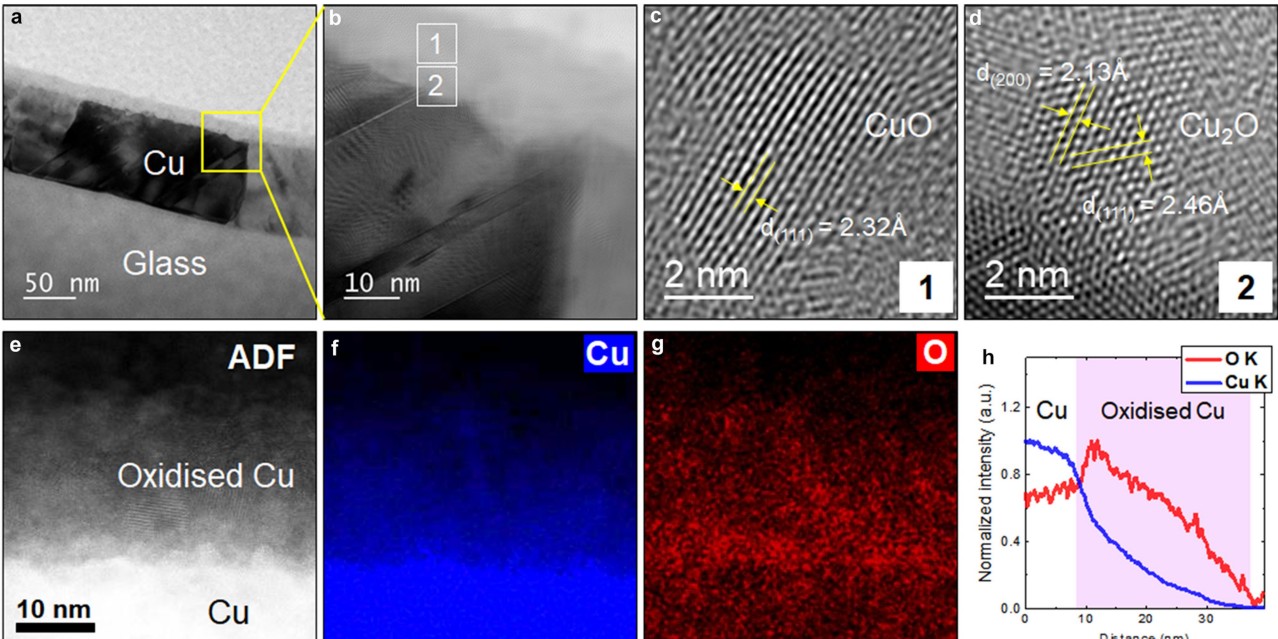

**Extended Data Fig. 2 | Structural and chemical characterizations of a conventional PCCF grown on a glass substrate. a**, Low-magnification cross-sectional BF-TEM image of the PCCF sample showing a typical polycrystalline structure. **b**, Magnified image of the region marked by the yellow box in **a**, which shows the presence of a thick oxidized Cu layer on the surface of the PCCF. **c**, **d**, HRTEM images of oxidized Cu nanoparticles formed in the regions denoted by white boxes in **b**. The measured planar spacings of the (111) planes are 2.32 and 2.46 Å, which represent typical values of CuO and Cu$_2$O phases, respectively. **e**, ADF-STEM image of the surface region of the PCCF. **f**, **g**, STEM-EDX elemental maps of Cu (**f**) and O (**g**) in the surface region. **h**, Vertically averaged intensity profiles of Cu (blue) and O (red) across the surface. Note that the intensity of each element was normalized with respect to its intensity maximum. From this result, it is evident that the highly oxidized Cu surface layer (indicated by the pink shaded region in the graph) can be formed on the PCCF with a thickness of tens of nanometres.

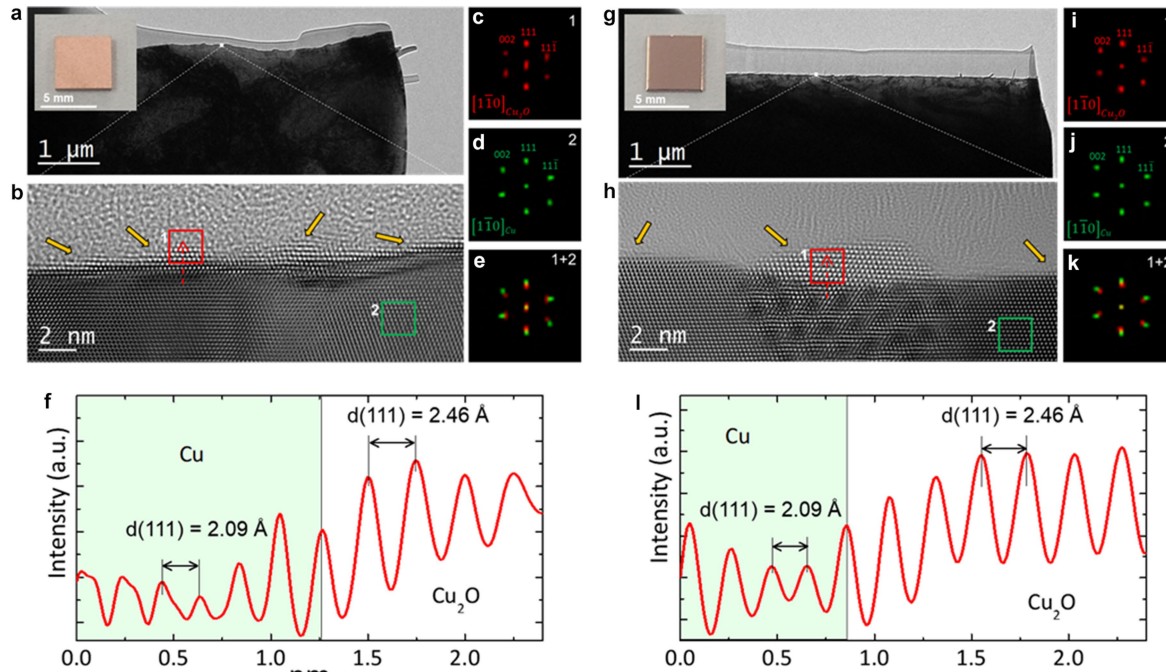

**Extended Data Fig. 3 | Surface oxidation of bulk Cu single crystal.**
**a**, **g**, Low-magnification micrographs of the non-polished and polished Cu single-crystal samples prepared by FIB milling. The insets show the two Cu single crystals sliced and cut to dimensions of $5 \times 5$ mm$^2$. **b**, **h**, HRTEM images of the two Cu surfaces. Note that the yellow arrows indicate the thin $Cu_2O$ layers that formed on the two surfaces. **c**, **d**, **i**, **j**, FFT patterns of the two regions of $Cu_2O$ (red box, labelled '1') and Cu (green box, labelled '2') depicted in each HRTEM image. **e**, **k**, Composite of the two FFT patterns for comparison. **f**, **l**, Measurements of the layer spacings across the interface between Cu and $Cu_2O$ (red dotted arrows).

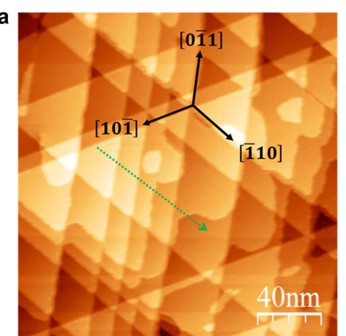

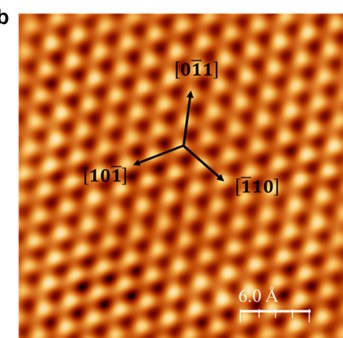

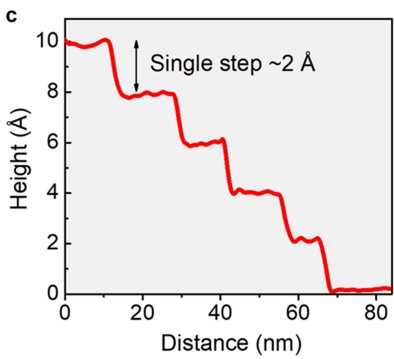

**Extended Data Fig. 4 | STM topography images of the clean Cu(111) single-crystal film at room temperature. a**, Configuration of step edges for an imaging area of 200 × 200 nm². **b**, Atomic-resolution surface structure (image size around 3 × 3 nm²). The arrows indicate the three corresponding <110> orientations on the (111) plane. **c**, Profile of step height obtained from the line scan marked with a black arrow in **a**, which shows a series of mono-atomic steps corresponding to the planar spacing of the Cu(111).

**Article**

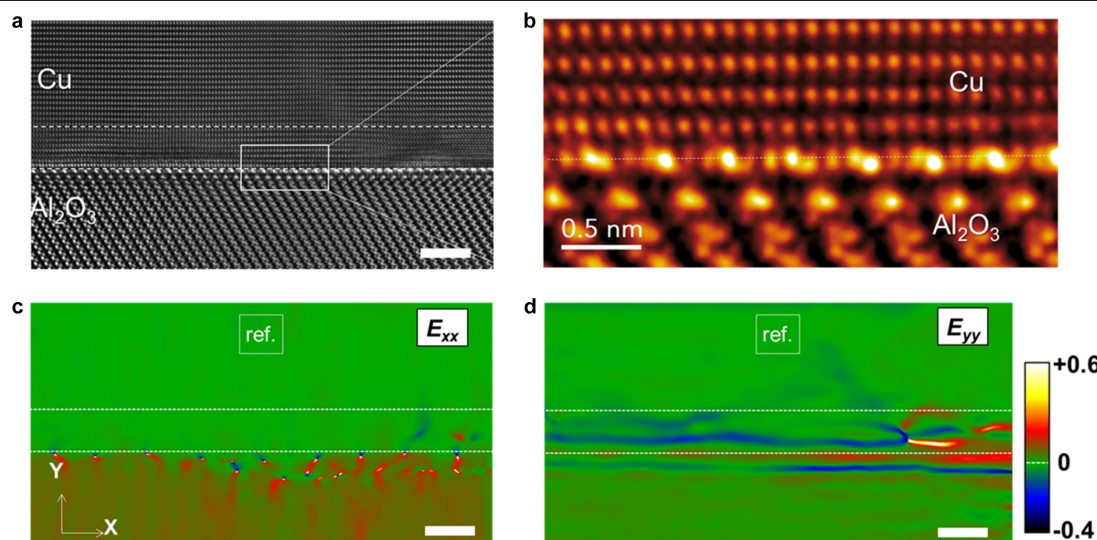

Scale bars: 2 nm.

**Extended Data Fig. 5 | Interface structure and strain distribution.**
**a**, Cross-sectional HRTEM image of the interface region of the Cu–$Al_2O_3$ heterostructure with an orientation relation of $(111)_{Cu}[11\bar{2}]_{Cu}//(001)_{Al_2O_3}[110]_{Al_2O_3}$. **b**, Enlarged image showing an abrupt interface structure. **c**, **d**, In-plane ($E_{xx}$) and out-of-plane ($E_{yy}$) strain field maps obtained by GPA of the HRTEM image. Note that the region enclosed by the white dotted line indicates the strained interface region of about 1–2 nm in thickness. The colour scale indicates the magnitude of the strain relative to the reference area marked by the white box. Scale bars are 2 nm.

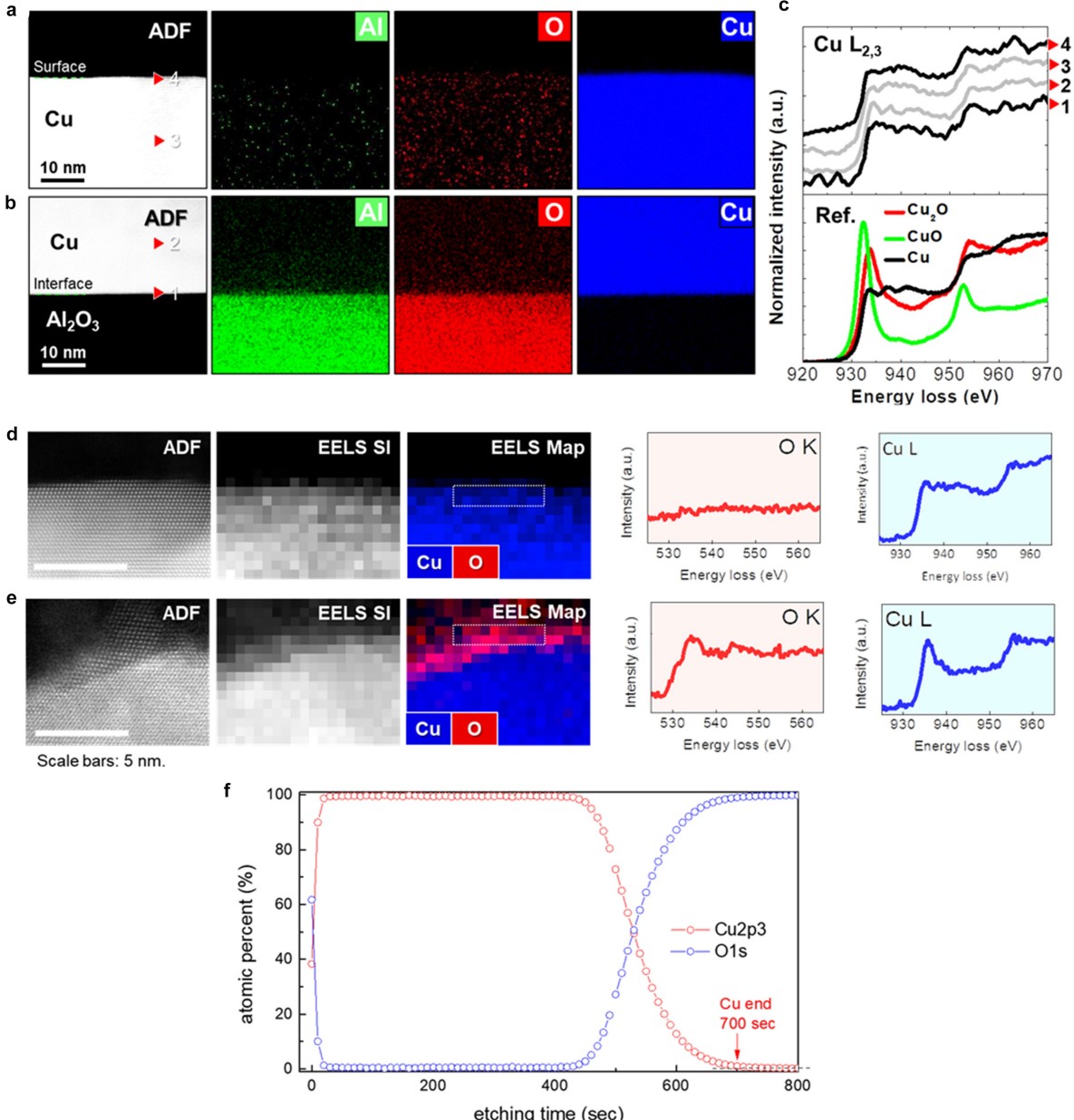

**Extended Data Fig. 6 | Chemical analysis and electronic structure of the grown SCCF. a**, **b**, Nanoscale STEM-EDX elemental maps of Al, O and Cu in the regions on the surface and at the interface of the Cu–Al$_2$O$_3$ heterostructure, respectively. **c**, Comparison of the Cu $L_{2,3}$ electron energy-loss near-edge structure profiles obtained at four different positions (marked by the numbers 1–4 in the ADF-STEM images) and the reference profiles obtained from copper metal and copper oxides, CuO and Cu$_2$O (lower graphs). **d**, ADF-STEM image, core-loss EELS spectrum imaging data and the constructed elemental map of Cu and O for the surface region of the SCCF sample. The red and blue profiles on the right side are core-loss EEL spectra of O $K$ and Cu $L_{2,3}$ edges extracted from the white dotted rectangle denoted in the EELS map. **e**, Equivalent data arrangement for the PCCF sample for comparison of surface oxidation. **f**, X-ray photoelectron spectroscopy depth profile depending on the etching time for the SCCF sample.

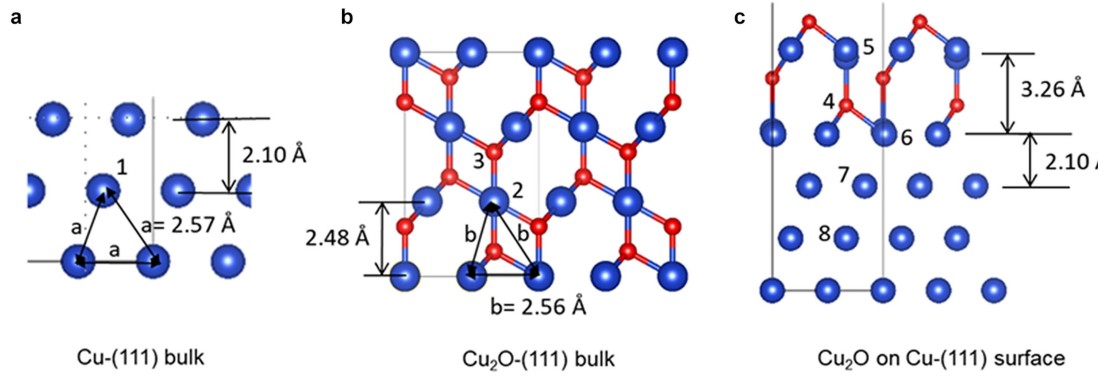

**d**

| | Cu bulk | Cu$_2$O bulk | Cu$_2$O monolayer on Cu(111) surface |
|---|---|---|---|
| Interlayer distance (Å) | 2.10 | 2.48 | 3.26 |
| Volume per Cu atom (Å$^3$) | 11.94 | 14.06 | 18.51 |

**e**

| System | Atom | $n_e$ | $n_0$ | $\delta$ |
|---|---|---|---|---|
| Cu bulk | Cu-1 | 11.00 | 11 | 0.00 |
| Cu$_2$O bulk | Cu-2 | 10.41 | 11 | +0.59 |
| | O-3 | 7.17 | 6 | -1.17 |
| Cu$_2$O monolayer on Cu(111) surface | O-4 | 7.09 | 6 | -1.09 |
| | Cu-5 | 10.42 | 11 | +0.58 |
| | Cu-6 | 10.74 | 11 | +0.26 |
| | Cu-7 | 10.99 | 11 | +0.01 |
| | Cu-8 | 11.00 | 11 | 0.00 |

**Extended Data Fig. 7 | Structural and electronic properties of Cu, Cu$_2$O and Cu$_2$O monolayers on the Cu(111) surface. a–c**, Structural model of Cu, Cu$_2$O and Cu$_2$O monolayers on the Cu(111) surface. Cu and O atoms are represented by blue and red spheres, respectively. Only the bonds between O and Cu atoms are drawn explicitly for clarity. **d**, Structural parameters of Cu, Cu$_2$O and Cu$_2$O monolayers on the Cu(111) surface. **e**, Degree of oxidation of atoms in various configurations. $n_e$ is the electron charge of each atom (the number of valence electrons attributed to each atom computed using the Bader decomposition method[43]) and $n_0$ is the number of valence electrons of the corresponding isolated neutral atom.

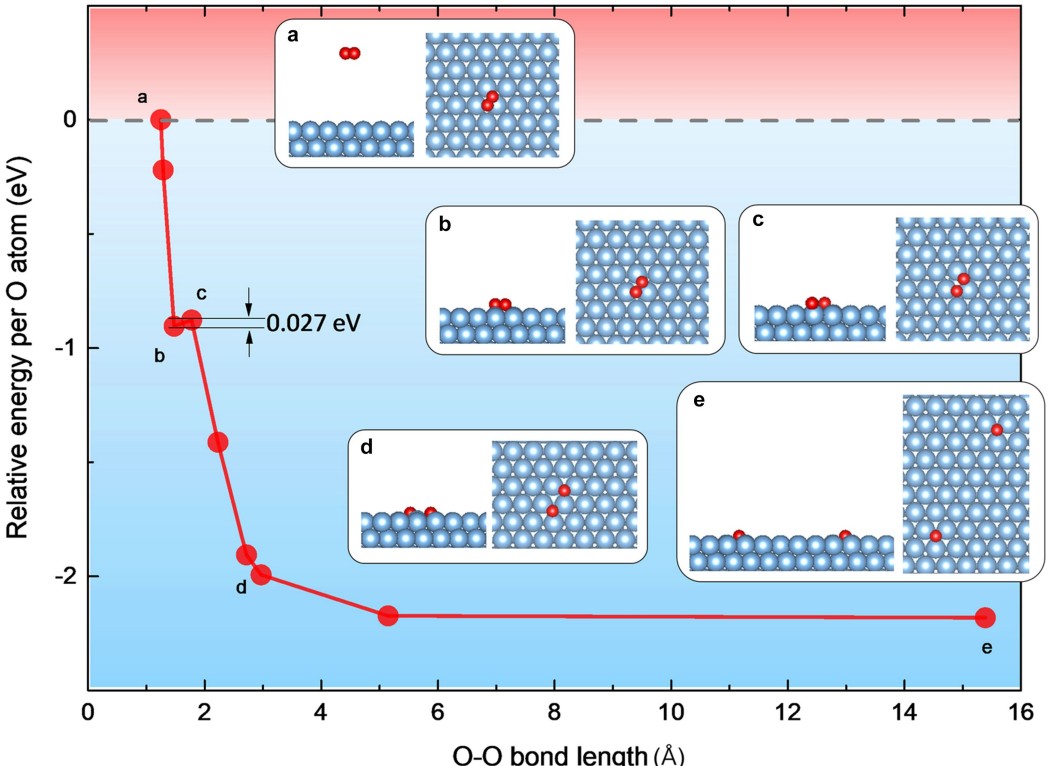

**Extended Data Fig. 8 | Relative energy per oxygen atom for the dissociation of the O₂ molecule approaching the Cu(111) surface as a function of O–O bond length.** Blue spheres represent Cu atoms and red spheres represent O atoms. The insets show the side and top views of the configurations, with corresponding letters on the curve. **a**, O₂ molecule far away from the surface. **b**, O₂ molecule in the physisorbed state. **c**, O₂ molecule in the transition state. **d**, Two O atoms adsorbed on the surface. **e**, O atoms diffused on the surface away from each other.

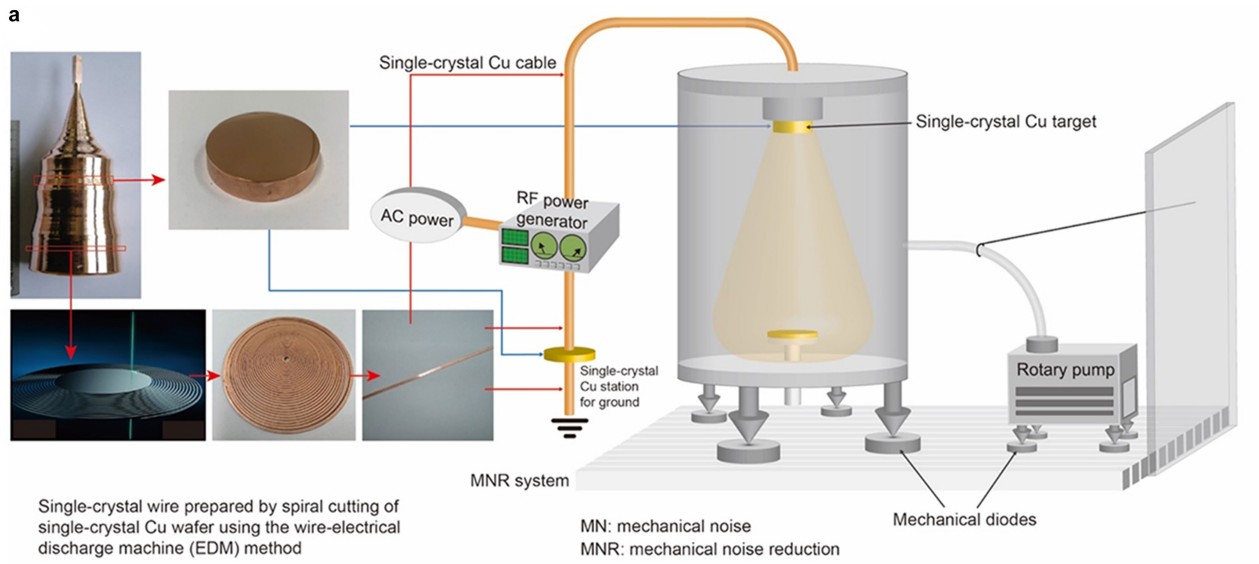

**a**

Single-crystal Cu cable

Single-crystal Cu target

AC power

RF power generator

Single-crystal Cu station for ground

Rotary pump

MNR system

MN: mechanical noise
MNR: mechanical noise reduction

Mechanical diodes

Single-crystal wire prepared by spiral cutting of single-crystal Cu wafer using the wire-electrical discharge machine (EDM) method

**b**

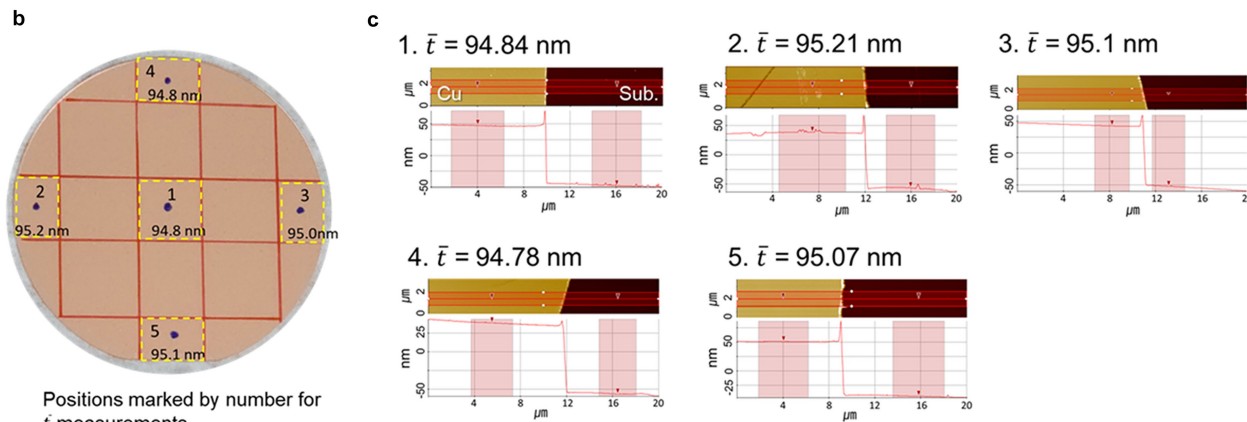

Positions marked by number for $\bar{t}$ measurements

**Extended Data Fig. 9 | Structure of the ASE system and uniformity of the grown Cu film. a**, Schematic diagram of the construction of the ASE system used in this study. The three key technological modifications made to differ from a commercially available sputtering system are as follows: single-crystal Cu sputtering target, electrical wiring with single-crystal Cu cables and mechanical noise-reduction parts, such as vibration absorbers and mechanical diodes. **b**, Two-inch SCCF grown wafer with a target thickness of 95 nm. To check the uniformity of the film thickness, the wafer was cut into pieces along the red lines and the five pieces marked by numbers were mounted to our AFM instrument to measure their thickness. **c**, Thickness profiles of the five samples. One side of each SCCF film was mechanically removed to expose the bare surface of the $Al_2O_3$ substrate to measure the film thickness from AFM edge profiling. Note that the mid-value in the thickness profile was set to be zero in AFM. The average thickness of the five samples was estimated to be $95 \pm 0.18$ nm, which corresponds to about 99.8% in uniformity.

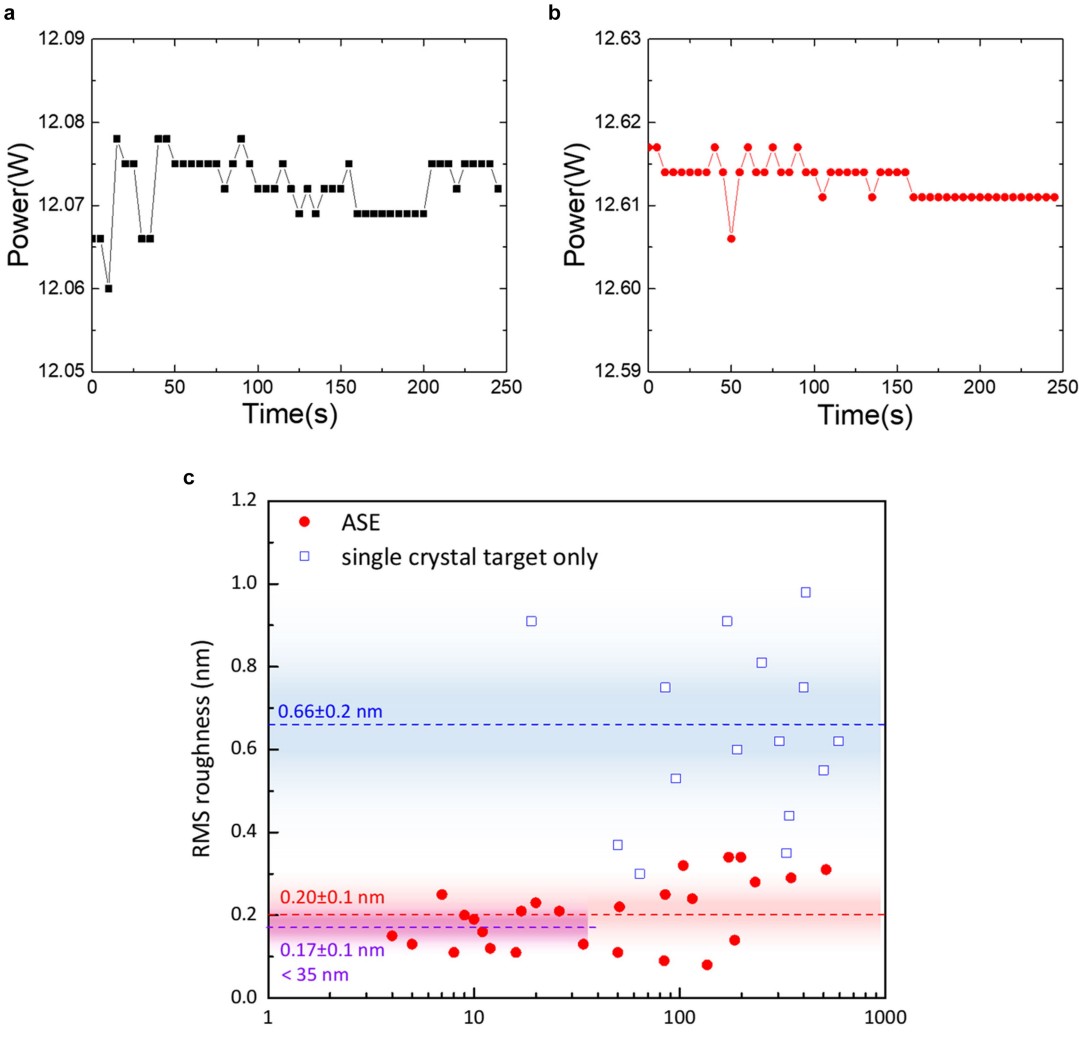

**Extended data Fig. 10 | Monitoring RF power stability and the effect of ASE on surface roughness. a**, **b**, Plots showing the output RF power fluctuations measured over time in the sputtering system without (**a**, black) and with a single-crystal Cu power cable (**b**, red). **c**, Plot of the measured root mean square (RMS) roughness of many samples as a function of the film thickness. The RMS values of the blue squares and the red circles were measured from the samples grown by the conventional sputtering system equipped with a single-crystal Cu target only and those grown using the ASE system, respectively. The thickness-dependent average RMS surface roughness of the blue squares (blue dashed line) and the red circles (red dashed line, ASE) are 0.66 ± 0.2 nm and 0.20 ± 0.1 nm, respectively. The average RMS surface roughness of the single-crystal films grown by the ASE method for <35 nm (purple dashed line) is 0.17 ± 0.1 nm.