## [Peer Review File · Nature]

Manuscript Title: Flat-surface-assisted and self-regulated oxidation resistance of Cu(111)

Reviewer Comments & Author Rebuttals

Reviewer Reports on the Initial Version:

Referee #2 (Remarks to the Author):

This paper reports a thorough study of the crystallographic surface structure of Cu thin films prepared by ASE (atomic sputtering epitaxy). Based on the results obtained in this analysis, authors conclude that pristine copper surfaces with very little concentration of surface steps can withstand long exposure times in air without undergoing a significant oxidation. This conclusion is reached after a careful study by TEM and electron diffraction of three sample situation, the pristine surfaces, the interface between the deposited copper and the alumina substrate and the surface after exposure to air for approximately one year. These experimental results are complemented with a theoretical analysis of the oxidation energetics.

Various key features stem from this study, the most important one related with the long time stability of the copper surfaces being the fact that surface defects are the natural way of enabling surface oxidation. From a technical point of view there are no substantial critics to this result, which is considered interesting. In fact, this is what one would have expected from common knowledge about metal surface oxidation. However, the most relevant feature of this work would have been the demonstration of a general procedure to get this defect-free copper surface. There are some concerns in this regard because there are not provided enough experimental details ensuring reproducibility in other laboratories, neither a discussion of the factors contributing to the flatness and densification of the films. Copper thin films are prepared by the ASE (atomic sputtering epitaxy) techniques and a few experimental details are provided about its realization as, for example, the use of single crystalline target or the implantation of a "mechanical noise reduction" system (no details provided). However, information is missing about critical parameters such as operating pressure, applied voltage and RF frequency, distance target-substrate and others of similar nature. All these parameters have a critical influence on the thermalization degree of incoming copper atoms, their trajectories and the possibility to accommodate them in compact structures once they reach the surface. Possible thermal contributions during the deposition process are neither considered or checked.

In summary, the paper is interesting, well written and carefully carried out from a technical point of view, particularly with regard to the crystal and surface state analysis and its complementary theoretical evaluation. However, the obtained results are within the knowing mechanisms of surface oxidation. Thus, the most interesting feature of the work, i.e., the development of a reproducible procedure to obtain very flat copper surfaces almost free from steps, is not sufficiently well explained, without practically any substantial detail concerning the thin film growth mechanism under the chosen experimental conditions. Clearly, this point is critical for reproducibility and the lack of information about it reduces credit for a general implementation of the reported results.

Referee #3 (Remarks to the Author):

This fascinating manuscript shows that Cu(111) is remarkably resistant to oxidation by molecular oxygen (over many years) when the surface is ultra-flat with no atomic steps or only monoatomic steps, and proposes an explanation for this based on computational simulation. In my opinion the experimental and computational results provide very robust and convincing evidence for the conclusions drawn. The importance of this work for is obvious given the many current and emerging applications of copper, particularly nanostructured copper. Consequently, I believe it will

be of interest to a broad audience and warrants publication in Nature. I have only two points that the authors might consider, both of which relate to how their remarkable findings connect to practical application. The authors can decide whether or not to address these in the manuscript.

1. If Cu₂O forms on the surface of Cu(111) due to diffusion of Cu from areas where there are multiple step edges followed by reaction with oxygen to form Cu₂O, does this not mean that polycrystalline Cu(111) will not oxidise if the surface is buried beneath another material that plugs the boundaries between crystallites (i.e. blocks diffusion of Cu from the multi-step edges)? This would explain why the slab-like 70 atom thick Cu films fabricated by Bellchambers et al.

<https://doi.org/10.3389/fmats.2018.00071>

proved to be remarkably stable in ambient air, even though they were perforated with millions of tiny apertures per square cm: The sheet resistance increased by <3.5% increase in sheet resistance after 7000 hrs in ambient air.

2. Ambient air contains water, and various sulphur containing compounds which are known to react with Cu. Could the authors comment on how, if at all the presence of these other compounds might (if at all) change the conclusions.

Author Rebuttals to Initial Comments:

Responses to the referees' comments:

We appreciate all the referees for reviewing our manuscript and providing valuable suggestions and advice to strengthen it. We have revised our manuscript in response to the referees' comments and addressed them point-by-point in this response letter. The changes made in the manuscript are indicated in red.

Referee #2 (Remarks to the Author):

This paper reports a thorough study of the crystallographic surface structure of Cu thin films prepared by ASE (atomic sputtering epitaxy). Based on the results obtained in this analysis, authors conclude that pristine copper surfaces with very little concentration of surface steps can withstand long exposure times in air without undergoing a significant oxidation. This conclusion is reached after a careful study by TEM and electron diffraction of three sample situation, the pristine surfaces, the interface between the deposited copper and the alumina substrate and the surface after exposure to air for approximately one year. These experimental results are complemented with a theoretical analysis of the oxidation energetics. Various key features stem from this study, the most important one related with the long time stability of the copper surfaces being the fact that surface defects are the natural way of enabling surface oxidation. From a technical point of view there are no substantial critics to this result, which is considered interesting. In fact, this is what one would have expected from common knowledge about metal surface oxidation.

Reply: We thank the referee for the encouraging remarks and helpful comments on our work. Here, we have made point-to-point responses to the referee's comments with supporting data below. We sincerely hope that the referee finds our responses satisfactory.

However, the most relevant feature of this work would have been the demonstration of a general procedure to get this defect-free copper surface. There are some concerns in this regard because there are not provided enough experimental details ensuring reproducibility in other laboratories, neither a discussion of the factors contributing to the flatness and densification of the films.

Reply: As noted by the referee, we agree that the reproducibility of the flat defect-free copper surface is important in this study; this requires a careful instrumental setup that is different from the conventional sputtering system. Hence, we describe a detailed configuration of the atomic sputtering epitaxy (ASE) system and the processing parameters to be considered. Because the system used in this study is constructed from reliable and readily available components, we anticipate that our

methodology can be exploited as a robust research platform for controlled metal surfaces and their related physical properties.

Before discussing the details of the ASE, we briefly summarise the key points for realising high reproducibility. Because the ASE system for controlling atomic-level growth is very sensitive to environmental factors, it requires three key instrumental modifications compared to a commercially available sputtering system.

First, we used a single-crystal target instead of a polycrystalline one, as mentioned in the manuscript; second, all conducting wires of the wiring network were replaced with single-crystal copper wires. Finally, a mechanical noise reduction system was installed to suppress the mechanical vibration coming from the surroundings. Once appropriately constructed, we confirmed that the quality of the Cu films was remarkably improved with a high level of reproducibility. Note that single-crystal copper is commercially available, and requests for the non-commercial use of the single-crystal wires and components can be made to the corresponding authors.

The schematic diagram for the ASE system with the three key features is given in **Figure R1**, and the detailed descriptions are as follows.

Figure R1. Schematic diagram of the construction of the ASE system used in this study. The three key technological modifications made from a commercially available sputtering system are featured: 1. single crystal Cu sputtering target, 2. electrical wiring with single crystal Cu cables, and 3. mechanical noise reduction parts such as vibration absorber and mechanical diodes.

1. Single crystal Cu sputtering target

For a polycrystalline target with a high density of surface steps and grain boundaries, atoms on the

edges or grain boundaries have a lower binding energy than that of atoms arranged on a flat plane of the grain. Hence, the atoms at the structural defects are prone to being sputtered as atomic clusters for the RF power set to remove them from a flat plane, which will eventually be deposited on the substrate as randomly oriented clusters. To enable the practicability of ASE growth, the use of a single-crystal Cu target with a (111) surface is essential because the Cu atoms are sputtered from the target as individual atoms. Thus, the uniform stacking of Cu atoms on the substrate is empirically achieved for the growth of an ultraflat film. The single-crystal Cu(111) target can be obtained from a single-crystal Cu ingot grown by the Czochralski method by cutting via wire-electrical discharge machining (wire-EDM), as a 6-mm-thick disc with a 2-inch diameter (see the two photos on the upper left side of Figure R1). Even though Cu single-crystal ingots are commercially available, we grew them using our own apparatus in this study, and there were no differences between the two in the resulting improvement of film quality.

2. Electrical noise reduction using single crystal Cu wires

To reduce electrical noise interference, we replaced the electrical networks made of conducting wires in conventional sputtering systems with single-crystal Cu wires as much as possible (see the three photos on the lower left side of Figure R1). The effectiveness of this modification was previously demonstrated in a Hall measurement kit with circuitry and connecting components made of single-crystal Cu, significantly improving the measurement precision of the electrical coefficients such as carrier density and mobility (Cha *et al.*, *Rev. Sci. Instrum.* **83** (2012) 013901). Single-crystal Cu wires can be prepared by the wire-EDM cutting of a single-crystal Cu disc in a spiral manner, as reported in our previous study (Cho *et al.*, *Crystal growth & Design* **10** (2010) 2780). To ensure further reduction of electrical noise, we replaced a typical RF power cable with a single crystal one. To monitor how effectively the single-crystal power cable improves the RF power stability in the sputtering system, we measured the change in the RF power over time with and without the single-crystal Cu power cable (Figure R2). Note that the single-crystal wiring has already been adapted in the sputtering system used in the test. It is evident that the output RF power is more stabilised within a narrower power range (12.61 ± 0.005 W) after changing the original power cable to a single crystal. Although the RF power stability of the sputtering system before the modification is also of good quality compared to that of the conventional ones, we empirically confirmed that such a level of stability is not sufficient for the growth of the Cu film with an atomic-level flatness.

Figure R2. Monitoring RF power stabilities. Plots showing the output RF power fluctuations measured over time in the sputtering system (a) without and (b) with a single crystal Cu power cable.

3. Mechanical noise reduction using mechanical diodes.

Although mechanical noise is not the main source of interference in the ASE system, the electrical noise reduction based on single-crystal Cu wiring would not be effective unless the mechanical noise is effectively screened. After the tests to reduce the mechanical noise with several choices including absorbers, barriers, vibration isolators, and vibration dampers, we found that the application of a mechanical diode consisting of a set of metal spikes and pads is very effective and economical to shield the mechanical interferences transmitted through the wall and floor. As depicted in Figure R1, we designed mechanical diodes and installed them on every device, including the chamber and vacuum pumps. The growth of atomically flat metal films was not successful without this shield against the mechanical noise.

To verify the reproducibility of our ASE approach for the growth of the ultraflat Cu(111) film, we measured the root mean square (RMS) roughness of many samples as a function of film thickness. Subsequently, we compared the values with those of the samples grown by a conventional sputtering system equipped only with the single-crystal Cu target (Figure R3). The averaged RMS value for the 29 samples grown by the ASE system was estimated to be $\sim 0.20 \pm 0.1$ nm (red dashed line), which is similar to the planar spacing of Cu(111). Notably, this can be further decreased to $\sim 0.17 \pm 0.1$ nm (purple dashed line) when considering thinner Cu films below 35 nm, suggesting their reliable applications to ultrathin electronic devices. However, the averaged RMS value for the samples grown by the conventional sputtering system equipped with the single-crystal Cu target was $\sim 0.66 \pm 0.2$ nm, which is good but not enough for the growth of the ultraflat Cu films only with the monoatomic steps.

Figure R3. Plot of the measured root mean square (RMS) roughness of many samples as a function of the film thickness. The RMS values of the blue squares and red circles were measured from the samples grown by the conventional sputtering system equipped with a single-crystal Cu target and those by the ASE system, respectively.

Copper thin films are prepared by the ASE (atomic sputtering epitaxy) techniques and a few experimental details are provided about its realization as, for example, the use of single crystalline target or the implantation of a “mechanical noise reduction” system (no details provided). However, information is missing about critical parameters such as operating pressure, applied voltage and RF frequency, distance target-substrate and others of similar nature. All these parameters have a critical influence on the thermalization degree of incoming copper atoms, their trajectories and the possibility to accommodate them in compact structures once they reach the surface. Possible thermal contributions during the deposition process are neither considered or checked.

Reply: In the ASE system discussed above, the base pressure of the system was 2×10^{-7} torr, and the working pressure with a 50 sccm Ar gas flow was 5×10^{-3} torr. In addition, it was revealed that the purity of the Ar gas affected the quality of the films. Thus, we used high-quality Ar gas with a purity of 99.9999% (6N) to grow the Cu films with atomically flat surfaces. It is empirically confirmed that such

high-purity Ar gas helps to improve the long-term surface stability, hardness, and adhesion to the substrate. The optimised deposition temperature and RF power were approximately 170 °C and 30 W, respectively, and these varied slightly depending on the ASE system.

Specific values of the growth parameters such as operating pressure, applied voltage, RF frequency, and distance from the target to the substrate are specified for the reader's information in the Methods section.

In summary, the paper is interesting, well written and carefully carried out from a technical point of view, particularly with regard to the crystal and surface state analysis and its complementary theoretical evaluation. However, the obtained results are within the knowing mechanisms of surface oxidation. Thus, the most interesting feature of the work, i.e., the development of a reproducible procedure to obtain very flat copper surfaces almost free from steps, is not sufficiently well explained, without practically any substantial detail concerning the thin film growth mechanism under the chosen experimental conditions. Clearly, this point is critical for reproducibility and the lack of information about it reduces credit for a general implementation of the reported results.

Reply: We sincerely thank the referee for the stimulating remarks on our work. As advised by the referee, we have provided a detailed explanation of our ASE system and the related parameters, together with extended data verifying the reproducibility.

We have thoroughly investigated the growth mechanism as an independent research to uncover how the atomically flat Cu film can grow. We found that an ultrathin and atomically flat Cu film can be grown via coplanar epitaxial growth accompanying the mode transition of the island to layered growth in the very early growth stage. The paper regarding this intriguing result has been submitted to another journal. We hope that it will soon be published for readers who are interested in the growth mechanism of the nearly defect-free Cu surface from a fundamental perspective. The authors are willing to provide any materials or related information on the system upon reasonable request for confirmation or further research.

According to the referee's suggestion, a detailed description of the ASE system, its reproducibility, and the optimised growth parameters have been added in the Methods section as follows:

(Revised text written in red on Line 321)

Methods

The atomic sputtering epitaxy (ASE) technique for the preparation of single-crystal Cu thin films. Cu thin films were grown as nearly defect-free and grain boundary-free single crystals using the ASE technique, addressing the problems of conventional sputter systems¹⁶. **Because the ASE system controlling atomic-level**

growth is very sensitive to environmental factors, it requires three key instrumental modifications compared to a commercially available sputtering system. First, we used a single-crystal target instead of a polycrystalline one; second, all conducting wires of the wiring network were replaced with single-crystal copper wires. Finally, a mechanical noise reduction system was installed to suppress the mechanical vibrations from the surroundings. The idea of this technology is based on completely eliminating the noise caused by electron-grain boundary scattering in the conduction network within the device, which has been completely ignored until now. Once the system was appropriately constructed, we confirmed that the quality of the Cu films was remarkably improved with a high level of reproducibility. The schematic diagram of the ASE system with the three key features is provided in Extended Data Fig. 9, and the detailed descriptions are as follows.

1. Single-crystal Cu sputtering target

For a polycrystalline target with a high density of surface steps and grain boundaries, atoms on the edges or grain boundaries have a lower binding energy than that of atoms arranged on a flat plane of the grain. Hence, the atoms at the structural defects are prone to being sputtered as atomic clusters for the RF power set to remove atoms from a flat plane, which will eventually be deposited on the substrate as randomly oriented clusters. To realise the practicability of ASE growth, the use of a single-crystal Cu target with a (111) surface is essential because Cu atoms are sputtered from the target as individual atoms. Thus, the uniform stacking of Cu atoms on the substrate is empirically achieved for the growth of an ultraflat film. The single-crystal Cu(111) target can be obtained from the single-crystal Cu ingot grown by the Czochralski method by cutting via wire-electrical discharge machining (wire-EDM) as a 6-mm-thick disc with a 2-inch diameter (see the two images on the upper left side of Extended Data Fig. 9). Even though Cu single-crystal ingots are commercially available, we grew them using our own apparatus in this study, and there were no differences between the two in the resulting improvement of film quality.

2. Electrical noise reduction using single crystal Cu wires

To reduce electrical noise interference, we replaced the electrical networks made of conducting wires in conventional sputtering systems with single-crystal Cu wires as much as possible (see the three images on the lower left side of Extended Data Fig. 9). The effectiveness of this modification was previously demonstrated in a Hall measurement kit with circuitry and connecting components made of single-crystal Cu, significantly improving the measurement precision of the electrical coefficients such as carrier density and mobility³⁶. Single-crystal Cu wires can be prepared by the wire-EDM cutting of a single-crystal Cu disc in a spiral manner, as reported in our previous study³⁷. To ensure further reduction of electrical noise, we replaced a typical RF power cable with a single crystal. To monitor how effectively the single-crystal power cable improves the RF power stability in the sputtering system, we measured the change in the RF power over time with and without the single-crystal Cu power cable (Extended Data Fig. 10a, b). Single-crystal wiring has already been adapted in the sputtering system used in the test. Therefore, it is evident that the output RF power is more stabilised within a narrower power range (12.61 ± 0.005 W) after changing the original power cable to a single crystal. Although the RF power stability of the sputtering system before modification is also of good quality compared to the

conventional ones, we empirically confirmed that such a level of stability is not sufficient for the growth of the Cu film with an atomic-level flatness.

3. Mechanical noise reduction using mechanical diodes.

Although mechanical noise is not the main source of interference in the ASE system, the electrical noise reduction based on the single-crystal Cu wiring cannot be effective unless the mechanical noise is effectively screened. After the tests to reduce the mechanical noise with several choices including absorbers, barriers, vibration isolators, and vibration dampers, we found that the application of a mechanical diode consisting of a set of metal spikes and pads is very effective and economical in shielding the mechanical interferences transmitted through the wall and the floor. As depicted in Extended Data Fig. 9, we designed mechanical diodes and installed them on every device, including the chamber and vacuum pumps. The growth of the atomically flat metal films was not successful without this shield against mechanical noise.

To verify the reproducibility of our ASE approach for the growth of the ultraflat Cu(111) film, we measured the root mean square (RMS) roughness of many samples as a function of the film thickness and compared these values with those of the samples grown by a conventional sputtering system equipped only with the single-crystal Cu target (Extended data Fig. 10c). The averaged RMS value for the 29 samples grown by the ASE system was estimated to be $\sim 0.20 \pm 0.1$ nm (red dashed line), which is similar to the planar spacing of Cu(111). Notably, it can be further decreased to $\sim 0.17 \pm 0.1$ nm (purple dashed line) when considering thinner Cu films below 35 nm, suggesting their reliable applications to ultrathin electronic devices. However, the averaged RMS value for the samples grown by the conventional sputtering system equipped with the single-crystal Cu target was $\sim 0.66 \pm 0.2$ nm, which is good but not enough for the growth of the ultraflat Cu films only with the monoatomic steps.

Optimised sputtering conditions using the ASE system. A double-side polished (001) Al₂O₃ wafer with a thickness of 430 μ m was used as the substrate material. The optimised deposition temperature and RF (13.56 MHz in frequency) power were about 170 °C and 30 W, respectively, and varied slightly depending on the ASE systems. The target-to-substrate distance was set at 95 mm. The base pressure was maintained at under 2×10^{-7} torr, and the working pressure at 5×10^{-3} torr with an Ar gas flow of 50 sccm. Ar gas with a purity of 99.9999% (6N) was used. The relationship between the deposition time and the thickness of the thin film (or the average growth rate) was determined from the average deposition time of a 200-nm-thick film grown under the optimum conditions. The determined average growth rate of ~ 4.3 nm/min is fairly reliable above a film thickness of 10 nm.

Extended Data Fig. 9 | Schematic diagram of the construction of the ASE system used in this study. The three key technological modifications made to differ from a commercially available sputtering system are as follows: single-crystal Cu sputtering target, electrical wiring with single-crystal Cu cables, and mechanical noise reduction parts such as vibration absorbers and mechanical diodes.

Extended data Fig. 10 | Monitoring RF power stability and the effect of ASE on surface roughness. a, b, Plots showing the output RF power fluctuations measured over time in the sputtering

system (a, black) without and (b, red) with a single-crystal Cu power cable. **c**, Plot of the measured root mean square (RMS) roughness of many samples as a function of the film thickness. The RMS values of the blue squares and the red circles were measured from the samples grown by the conventional sputtering system equipped with a single-crystal Cu target only and those grown using the ASE system, respectively. The thickness-dependent average RMS surface roughness of the blue squares (blue dashed line) and the red circles (red dashed line, ASE) are 0.66 ± 0.2 nm and 0.20 ± 0.1 nm, respectively. The average RMS surface roughness of the single-crystal films grown by the ASE method for < 35 nm (purple dashed line) is 0.17 ± 0.1 nm.

Referee #3 (Remarks to the Author):

This fascinating manuscript shows that Cu(111) is remarkably resistant to oxidation by molecular oxygen (over many years) when the surface is ultra-flat with no atomic steps or only monoatomic steps, and proposes an explanation for this based on computational simulation. In my opinion the experimental and computational results provide very robust and convincing evidence for the conclusions drawn. The importance of this work for is obvious given the many current and emerging applications of copper, particularly nanostructured copper. Consequently, I believe it will be of interest to a broad audience and warrants publication in Nature. I have only two points that the authors might consider, both of which relate to how their remarkable findings connect to practical application. The authors can decide whether or not to address these in the manuscript.

Reply: We are grateful to the referee for the encouraging remarks and insightful comments on the possible expandability of our ultraflat Cu film for application as a super-corrosion-resistant material that could work in harsh and corrosive environments.

1. If Cu₂O forms on the surface of Cu(111) due to diffusion of Cu from areas where there are multiple step edges followed by reaction with oxygen to form Cu₂O, does this not mean that polycrystalline Cu(111) will not oxidise if the surface is buried beneath another material that plugs the boundaries between crystallites (i.e. blocks diffusion of Cu from the multi-step edges)? This would explain why the slab-like 70 atom thick Cu films fabricated by Bellchambers et al. <https://doi.org/10.3389/fmats.2018.00071> proved to be remarkably stable in ambient air, even though they were perforated with millions of tiny apertures per square cm: The sheet resistance increased by <3.5% increase in sheet resistance after 7000 hrs in ambient air.

Reply: We thank the referee for the comment that our theoretical discovery can probably elucidate the remarkable effectiveness of a thin Al overlayer at passivating the underlying polycrystalline Cu film. In the study mentioned above, it was revealed that the deposited Al on Cu is prone to segregate to the Cu grain boundaries, which are more susceptible to oxidation, in the Cu film surface to eventually form a self-limiting oxide. This very thin oxide can play the role of a surface flattener to reduce the overall film surface roughness because it preferentially fills in grooved surfaces located at the grain boundaries. It can also act as a shield to block the further progress of oxidation; a copper-doped aluminum oxide plug can serve as an effective local barrier to oxygen and moisture. This passivation effect was maintained stably even in the patterned Cu film.

As mentioned by the referee, we agree that the notable efficacy of the passivation approach could be understood by the effective screening of the diffusion of oxygen atoms to the multi-step edges

(highly exposed at the grain boundaries) from the perspective of an atomic-scale energy landscape. Once passivated, the diffusion barrier can effectively work even in the patterned Cu film for screening the exposure of multi-step edges to oxygen in the air. We appreciate the referee inspiring us for the field-widened perspective of our ASE approach and the relevant atomistic interpretation. To convey the implication of the effective passivation approach to the readership, we cited this study as an introductory background.

(Page 2, Line 45, revised) "... Although there are numerous studies on the oxidation of copper including practical passivation techniques,¹¹ the fact that a flat copper surface itself can be free of oxidation has been ignored, ...".

[11] Bellchambers, P., Lee, J., Varagnolo, S., Amari, H., Walker, M. & Hatton, R. A. Elucidating the exceptional passivation effect of 0.8 nm evaporated aluminum on transparent copper films. *Front. Mater.* **5**, 71, doi:10.3389/fmats.2018.00071 (2018).

2. Ambient air contains water, and various sulphur containing compounds which are known to react with Cu. Could the authors comment on how, if at all the presence of these other compounds might (if at all) change the conclusions.

Reply: We agree with the reviewer's insightful comment on the possible reactivity of the Cu(111) surface to other corrosive gases such as sulfur. In addition to its potential applications as nanoelectronic devices that require an ultraflat surface and superb oxidation resistance of the copper film, we believe that the referee's advice hints that our Cu thin film with an ultraflat surface could be more widely used even in extreme environments filled with harsh and corrosive gases, provided it can still exhibit outstanding resistance. To determine the reactivity of the ultraflat Cu(111) surface with sulfur, we compared the Ellingham diagrams of Cu₂O and Cu₂S and conducted DFT calculations for the energy required for copper sulfurisation with the atomic model shown in Figure 4d by substituting S for O (Figure R4). From the Ellingham diagrams comparing the thermodynamic stabilities of copper oxides and copper sulfides (Figure R4a and b), we see that copper oxide is more stable than sulfur oxide at room temperature. This suggests that the sulfurisation of our ultraflat Cu(111) would require more energy than oxidation. This was verified by DFT modelling, as shown in Figure R4c. We found that sulfurisation is a typical endothermic reaction in monoatomic steps, as expected, and it requires a much larger activation energy for the penetration of an S atom through a Cu(111) surface compared to oxidation. In other words, the ultraflat Cu(111) surface with a low density of monoatomic steps would show a much stronger resistance to sulfur than oxygen. Given this result, we can therefore carefully assert that the conclusion of our manuscript is still valid in the presence of other types of gases. Even though it is beyond the scope of this study, we believe that this preliminary result showing

the outstanding chemical stability of the ultraflat Cu film can stimulate further systematic investigations in the presence of various corrosive gases (or liquids) or their compounds. We would like to thank the referee for the valuable comment that provided us with a more expandable research boundary than we envisioned.

Figure R4 (For referee-only) **a, b**, Comparison of the Ellingham diagrams of Cu_2O (red) and Cu_2S (green). The Gibbs free energies of formation of the oxide and sulfide at room temperature are marked in the plots with red and green diamonds, respectively. From the diagrams, it is evaluated that the Cu_2O is thermodynamically more stable than the Cu_2S at the temperature below 900 K. Note that data is adapted from “Thomas B. Reed, Free Energy of Formation of Binary Compounds, MIT Press, Cambridge, MA, 1971”. **c**, Comparison of energy profiles of the O atom (red, data shown in Fig. 4d) and the S atom (blue) along the penetration path from the outside to the inside of the Cu(111) surface.

As a minor Change,

We moved the ref.36~40 to the next of Data availability statements.

===== End of comments =====

Reviewer Reports on the First Revision:

Referee #2 (Remarks to the Author):

Although the admendements made in teh manuscript addresses most of the queries arisen after the first reading of the paper and, therefore, it might be considered almost ready for publication, there are still some minor points that might be addressed by authors in order to provide the reader with an important information.

As requested in the first revision, authors have provided a more detailed description of the experimental details regarding the sputtering process. However, these experimental details would gain a clearer meaning if there were accompanied by some estimation of the kinetic energy distribution of sputtered atoms and, possibly their angular distribution function. These magnitudes are dependent on working conditions (pressure, distance between target and substrate, etc.) and can be estimated using conventional models describing the magnetron sputtering models. Particularly, the kinetic energy of the incoming atoms can be very decisive in controlling the growth process of the films and would be a very important magnitude in order to justify the findings of the work. I advise to take into consideration this kinetic energy (usually around ten or more eVs) to justify the main findings of the work.

Referee #3 (Remarks to the Author):

I have carefully read the revised submission and am satisfied that the two points I raised have been addressed.

In my opinion the statement made by the other reviewer of this manuscript that the main findings are 'what one would have expected from common knowledge about metal surface oxidation' is overly harsh and not really justified. The current manuscript gives new, in depth fundamental insight into why atomically flat Cu surfaces and surfaces with monotomic steps are so stable towards oxidation in air, and shows that flat Cu surfaces can be oxidation free over a period of years - which (in my opinion) cannot be regarded as 'common knowledge'. Given the technological importance of Cu for numerous emerging devices this insight is very important and will motivate others to develop scalable ways to achieve this for practical applications. The authors have presented one such way, and in the revised manuscript have provided more experimental details, but I do not regard that protocol as the central/most important finding of this paper.

In my opinion the revised manuscript meets the very high standard needed for publication in Nature and will be of interest to a broad audience of scientists.

Author Rebuttals to First Revision:

Responses to the referees' comments:

We appreciate the referees for carefully reviewing our revised manuscript and response letter. We are deeply grateful for the referees for giving high ratings and encouraging comments on our manuscript. We have revised our manuscript in response to the referees' comments and addressed them point-by-point in this response letter. The changes made in the manuscript are indicated in red.

Referee #2 (Remarks to the Author):

Although the admendements made in the manuscript addresses most of the queries arisen

after the first reading of the paper and, therefore, it might be considered almost ready for publication, there are still some minor points that might be addressed by authors in order to provide the reader with an important information.

As requested in the first revision, authors have provided a more detailed description of the experimental details regarding the sputtering process. However, these experimental details would gain a clearer meaning if there were accompanied by some estimation of the kinetic energy distribution of sputtered atoms and, possibly their angular distribution function. These magnitudes are dependent on working conditions (pressure, distance between target and substrate, etc.) and can be estimated using conventional models describing the magnetron sputtering models. Particularly, the kinetic energy of the incoming atoms can be very decisive in controlling the growth process of the films and would be a very important magnitude in order to justify the findings of the work. I advise to take into consideration this kinetic energy (usually around ten or more eVs) to justify the main findings of the work.

Reply: We thank the referee for suggesting the publication of our manuscript and inspiring comments on our work, which make us look at our results from a different perspective. The referee put an emphasis on the knowledge of energetics of the sputtered copper atoms (kinetic energy of these atoms and their angular distribution function), which is an important information to understand how the ultraflat Cu surface grows. The kinetic energy of sputtered Cu atoms depends on the incident ion energy of Ar^+ and the binding energy of Cu atoms at the surface of the target. The crystallographically different surfaces of Cu have different surface binding energies ($E_{b, \text{Cu}}$), which were reported to be 4.62, 4.26, and 4.65 eV for Cu(100), Cu(100), and Cu(111) planes, respectively (Kudriavtsev, Y. et al., *Appl. Surf. Sci.* **239**, 273, 2005). Considering the potential energy ($E_{\text{Ar}^+} = 15$ eV) of the accelerated Ar^+ ion at the maximum current of 1 A in our atomic sputtering epitaxy (ASE) equipment, the kinetic energy of the sputtered Cu atoms can be narrowly distributed at around 10.35 eV, which is roughly calculated from $E_{\text{Ar}^+} - E_{b, \text{Cu}(111)}$ in the case of Cu(111) single-crystal target used in this study. In contrast, in the case of polycrystalline Cu target dominantly having a mixture of Cu(100), Cu(110), and Cu(111) exposed planes at the surface, the kinetic energy of the sputtered Cu atoms is expected to be spread between 10.35 eV and 10.74 eV. When considering the surface defects and grain boundaries at the polycrystalline target, the kinetic energy would be distributed more broadly up to 11.52 eV (Jackson D. P., *Radiation Effects* **18**(3-4), 185, 1973).

The radial distribution of the incident flux of the sputtered Cu atoms at the substrate is known to determine the uniformity of the deposition thickness of Cu film (Yagisawa, T. &

Makabe, T. *J. Vac. Sci. Technol. A* **24**, 908, 2006). To check the thickness uniformity of the grown SCCF, we measured the thickness at five different-points from centre to edge in a 2-inch wafer using an atomic force microscope (AFM). As a result, the thickness uniformity was estimated to be ~99.8% (see revised Extended Data Fig. 9b, c). This result suggests that the diffusive flux of Cu atoms is purely uniform at the position of the substrate. Note that the substrate of our system is rotated with 30 rpm.

To convey the implication of the transport behaviour of sputtered Cu atoms to influence the growth process, we added the above discussion in the 'Methods' section and cited the related papers as follows:

(Page 14, Line 391, revised) "...The kinetic energy of sputtered Cu atoms depends on the incident ion energy of Ar⁺ and the binding energy of Cu atoms at the surface of the target. The crystallographically different surfaces of Cu have different surface binding energies ($E_{b,Cu}$), which were reported to be 4.62, 4.26, and 4.65 eV for Cu(100), Cu(110), and Cu(111) planes, respectively³⁸. Considering the potential energy ($E_{Ar^+} = 15$ eV) of the accelerated Ar⁺ ion at the maximum current of 1 A in our ASE equipment, the kinetic energy of the sputtered Cu atoms can be narrowly distributed at around 10.35 eV, which is roughly calculated from $E_{Ar^+} - E_{b,Cu(111)}$ in the case of Cu(111) single-crystal target used in this study. In contrast, in the case of polycrystalline Cu target dominantly having a mixture of Cu(100), Cu(110), and Cu(111) exposed planes at the surface, the kinetic energy of the sputtered Cu atoms is expected to be spread between 10.35 and 10.74 eV. When considering the surface defects and grain boundaries at the target, the kinetic energy would be distributed more widely up to 11.52 eV³⁹. The radial distribution of the incident flux of the sputtered Cu atoms at the substrate is known to determine the uniformity of the deposition thickness of Cu film⁴⁰. To check the thickness uniformity of the grown SCCF, we measured the thickness at five different points from centre to edge in a 2-inch wafer using an atomic force microscope (AFM). As a result, the thickness uniformity was estimated to be ~99.8% (Extended Data Fig. 9b, c). This result suggests that the diffusive flux of Cu atoms is purely uniform at the position of the substrate. Note that the substrate of our system is rotated with 30 rpm."

38 Jackson, D. P. Binding energies in cubic metal surfaces. *Radiat. Eff.* **18**, 185-189, doi:10.1080/00337577308232120 (1973).

39 Kudriavtsev, Y., Villegas, A., Godines, A. & Asomoza, R. Calculation of the surface binding energy for ion sputtered particles. *Appl. Surf. Sci.* **239**, 273-278, doi:10.1016/j.apsusc.2004.06.014 (2005).

40 Yagisawa, T. & Makabe, T. Modeling of dc magnetron plasma for sputtering: Transport of sputtered copper atoms. *J. Vac. Sci. Technol. A* **24**, 908-913, doi:10.1116/1.2198866 (2006).

(Revised) Extended Data Fig. 9 | Structure of the ASE system and uniformity of the grown Cu film.

a, Schematic diagram of the construction of the ASE system used in this study. The three key technological modifications made to differ from a commercially available sputtering system are as follows: single-crystal Cu sputtering target, electrical wiring with single-crystal Cu cables, and

mechanical noise reduction parts such as vibration absorbers and mechanical diodes. **b**, 2-inch SCCF grown wafer with a target thickness of 95 nm. To check the uniformity of the film thickness, the wafer was cut into pieces along the red lines and the five pieces marked by numbers were mounted to our AFM instrument to measure their thickness. **c**, Thickness profiles of the five samples. One side of each SCCF film was mechanically removed to expose the bare surface of the Al_2O_3 substrate to measure the film thickness from AFM edge profiling. Note that the mid-value in the thickness profile was set to be zero in AFM. The average thickness of the five samples was estimated to be 95 ± 0.18 nm, which corresponds to $\sim 99.8\%$ in uniformity.

Referee #3 (Remarks to the Author):

I have carefully read the revised submission and am satisfied that the two points I raised have been addressed. In my opinion the statement made by the other reviewer of this manuscript that the main findings are 'what one would have expected from common knowledge about metal surface oxidation' is overly harsh and not really justified. The current manuscript gives new, in depth fundamental insight into why atomically flat Cu surfaces and surfaces with monoatomic steps are so stable towards oxidation in air, and shows that flat Cu surfaces can be oxidation free over a period of years - which (in my opinion) cannot be regarded as 'common knowledge'. Given the technological importance of Cu for numerous emerging devices this insight is very important and will motivate others to develop scalable ways to achieve this for practical applications. The authors have presented one such way, and in the revised manuscript have provided more experimental details, but I do not regard that protocol as the central/most important finding of this paper.

In my opinion the revised manuscript meets the very high standard needed for publication in Nature and will be of interest to a broad audience of scientists.

Reply: We are deeply grateful to Referee #3 for the encouraging comments and the high evaluation on our study. We believe that the referee's insightful advice given during the review process will stimulate us or others to move toward further research for novel applications using the developed ASE system.